# Current Status of Liquid Metal Printing

**Troy Y. Ansell** (ID)

Mechanical and Aerospace Engineering, Naval Postgraduate School, 700 Dyer Road, Monterey, CA 93943, USA; troy.ansell@nps.edu; Tel.: +1-831-656-3033

**Abstract:** This review focuses on the current state of the art in liquid metal additive manufacturing (AM), an emerging and growing family of related printing technologies used to fabricate near-net shape or fully free-standing metal objects. The various printing modes and droplet generation techniques as applied to liquid metals are discussed. Two different printing modes, continuous and drop-on-demand (DOD), exist for liquid metal printing and are based on commercial inkjet printing technology. Several techniques are in various stages of development from laboratory testing, prototyping, to full commercialization. Printing techniques include metal droplet generation by piezoelectric actuation or impact-driven, electrostatic, pneumatic, electrohydrodynamic (EHD), magnetohydrodynamic (MHD) ejection, or droplet generation by application of a high-power laser. The impetus for development of liquid metal printing was the precise, and often small scale, jetting of solder alloys for microelectronics applications. The fabrication of higher-melting-point metals and alloys and the printing of free-standing metal objects has provided further motivation for the research and development of liquid metal printing.

**Keywords:** additive manufacturing; ink-jet printing; liquid metal printing

## 1. Introduction

The world is currently witnessing a modern industrial revolution take place. Commentators often refer to this as the fourth industrial revolution or "Industry 4.0." As with the historical industrial revolutions of the 18th, 19th, and 20th centuries, multiple factors have contributed to the current disruption in industry and society [1]. Computers and the internet have revolutionized how we share information and how we interact with each other (often referred to as the third industrial revolution or "Industry 3.0" when combined with robotics). Further, the emergence of big data and artificial intelligence is changing how we interact with the world. With the manipulation of materials at the nanoscale and the integration of nanoparticles in macroscale parts, engineers are on the cusp of commercializing products made from "designer" materials with highly customizable properties [2–6]. Finally, innovative new manufacturing technologies, e.g., computer-aided design/computer-aided manufacturing (CAD/CAM) and automated production equipment, robotics, and 3D printing are changing how businesses either design, prototype, and produce consumer products or acquire components for a larger products [1]. Printers when combined with CAD/CAM enable fabrication of complex parts often beyond the ability of traditional casting methods including investment casting or particle injection molding techniques. Additive manufacturing (AM) is the term which broadly encompasses these disruptive technologies that are allowing everyone from large manufacturers down to the consumer the ability to print 3D (and 4D) objects.

Until recently, 3D printers have been restricted to prototyping parts; however, improvements in printer design, printer commercialization, and computer processing has enabled full part (or component) manufacturing [7,8]. Work by numerous scientists and engineers starting in the 1960s on through the present have advanced the various printing methods to a point where fully free-standing polymeric, ceramic, and/or metallic parts are printable. The current research focus in additive manufacturing includes the development

of multi-material printers; decreasing defects in printed parts; expanding the current list of printable materials, especially ceramics and non-weldable metallic materials; and lowering the cost of production through AM [9]. One exciting group of AM techniques involves the jetting of liquid metal, following a digital pattern, into a solid freeform shape: liquid metal printing.

A brief timeline of some of the major developments in this family of techniques is provided in Figure 1. Liquid metal printing could potentially achieve several of the current AM development thrusts. One, liquid metal printing simplifies the 3D metal printing process. Common 3D metal printing methods includes powder-bed fusion, powder-feed, wire-feed, or binder-jet-based methods. These techniques require either high-power lasers and inert environments or extensive post-processing, especially the binder-jet method which requires pyrolysis after material jetting and before sintering. Second, liquid metal printing may allow for printing of non-weldable metals and metal alloys as opposed to powder-bed methods such as selective laser sintering/melting [10]. Liquid metal printing is scalable, more difficult for powder-based or wire-based AM methods, as the jetting technologies are based on commercially established inkjet printers and the volume of the liquid metal is not a limiting factor. Melting of recycled metal and jetting of molten fluid maybe another advantage of a droplet technique. A recent paper on recycling of metal powder used in a powder-bed fusion system highlighted the importance of reusing processed powder to lower the cost of AM parts [11]. Oxidation of the recycled powders affected the quality. Using an appropriate droplet technique may mitigate this effect. Finally, liquid metal printing expands the range of printable metal alloys as the only requirement is the ability to melt the material pre-jetting as opposed to melting or fusing powder either in situ or during post-processing.

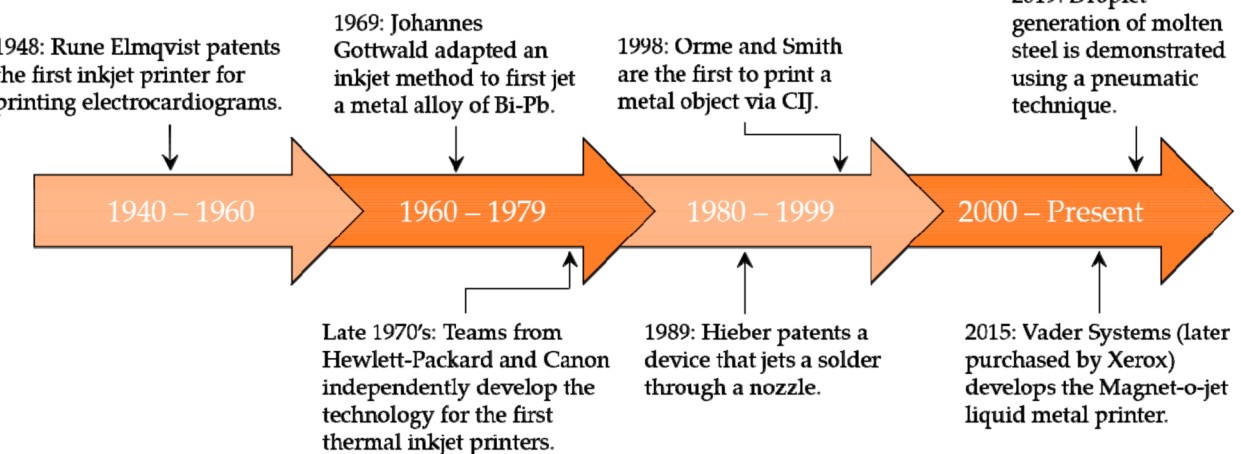

**Figure 1.** Timeline of some the important events in the history of liquid metal additive manufacturing. The early history is centered on the development of inkjet printing for 2D facsimiles of digital or analog (as is the case for the printing of electrocardiograms) information.

This review will focus on this beamless AM modality where free-standing 3D metal parts are fabricated and where the precursor is made of liquid (or molten) metal and not an ink loaded with metal particles. 3D printing techniques are split up into families of beam-based and beamless printing as seen in Figure 2. Powder bed fusion and electron beam fabrication, two of the more common printing methods, fall within the beam-based group of techniques often requiring a laser, electron beam, or another directed-energy pulse. Sheet lamination, binder jetting, material jetting, material extrusion, vat photo-polymerization constitute the group of beamless AM techniques. Liquid metal jet printing (highlighted in red in Figure 2) falls under the material jetting family of AM techniques. Material jetting is defined in the recent International Organization for Standardization/ASTM International (ISO/ASTM) standard on AM terminology as a "process in which

droplets of build material are selectively deposited" onto a substrate [12]. What follows is a background on additive manufacturing followed by a background discussion of inkjet printing and how the same technology used for printing words and pictures onto paper is also being used for the free-form fabrication of metal parts. Section 3 will cover liquid metal printing via continuous inkjet printing (CIJ). Section 4 will go into piezoelectric driven drop-on-demand (DOD) liquid metal printing. Section 5 will cover several field-induced DOD techniques. Sections 6 and 7 focuses on Pneumatic- and impact-driven DOD techniques, respectively. Section 8 will go into laser-induced DOD. An overview of printing considerations, future direction of this printing modality, and a conclusion will conclude this review.

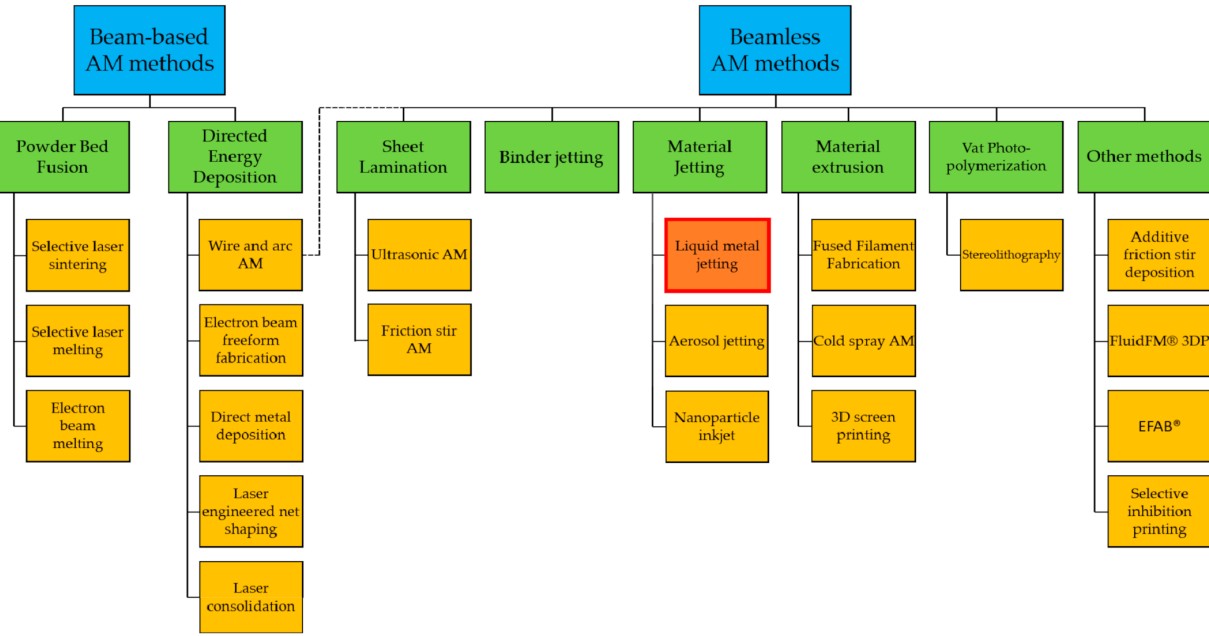

**Figure 2.** Taxonomy of additive manufacturing techniques. This review is focused on liquid metal jetting techniques highlighted in red. This figure is based on ISO/ASTM standard on additive manufacturing (AM) terminology [12].

## 2. Background

### 2.1. Brief Review of 3D Metal Printing

Polymers were among the first materials to be AM fabricated. Two of the early polymer printing technologies include stereolithography and fused deposition modeling or fused filament fabrication (FFF). Stereolithography was developed in Japan, the United States, and Europe independently in the early 1980s as a means of rapid prototyping of plastics via vat photopolymerization [13–19]. FFF was developed in the late 1980s by S. Scott Crump of Stratasys Inc. [20]. The most common polymer printing method is FFF which is also is the most common 3D printing technique in general. Other techniques have been developed for polymer and ceramic additive manufacturing [16,21–31]. The review by Zhangwei Chen et al. is an excellent and comprehensive overview of 3D printing of ceramics [32]. Functional composite materials, e.g., piezoelectric ceramic composites, have been printed via several methods including a $BaTiO_3$-based ceramic composite via digital light processing (a technique similar to stereolithography) [33]; a polyvinylidene fluoride composite loaded with $BaTiO_3$ nanoparticles and carbon nanotubes printed through FFF [34]; and $Pb(Zr_xTi_{1-x})O_3$ (PZT) ceramics by the selective laser sintering (SLS) method [35]. Selective laser sintering/melting (SLS/SLM) is a popular method of printing ceramic components. Similarly, for metals, SLS/SLM is a common choice of technique.

The 3D printing of metals is dominated by powder-based methods [9]. The dominate methods of 3D metal printing include laser powder bed fusion (LPBF) via a high-power

laser, i.e., SLS or SLM, and electron-beam powder bed fusion (EPBF). LPBF printers lay down a bed of atomized powder. A high-power pulsed laser either sinters or melts a pattern in the powder following a CAD pattern. Once a layer is complete, LPBF printers will lay down the next layer of powder and the process repeats until a full part is printed. The SLS and SLM techniques are similar, the main difference is sintering of powders occurs in SLS while full melting takes place in SLM. Electron-beam melting is like LPBF in that metal (or ceramic) powders are pre-alloyed, deposited, and fused in a layer-wise fashion. Of course, a major difference is the use of an electron beam to melt the powders as opposed to a laser. Although SLS and SLM are commonly performed in an inert atmosphere or even a low vacuum environment, this is not a requirement. In EPBF, a high vacuum (at least in the range of $10^{-5}$ mbar) is a requirement as an electron beam will be scattered by atmospheric gases.

These powder-based AM techniques generally lead to materials with unique microstructures. Example images of the titanium alloy, Ti-6Al-4V, are shown in Figure 3. This titanium alloy, depending on processing, develops a mixture of lamella grains ($\alpha$ + Ti$_3$Al), equiaxed and columnar $\alpha$-Ti and intergranular $\beta$-Ti as seen in Figure 3a [36,37]. When this alloy is fabricated into a part via SLM, a martensitic acicular $\alpha$ phase develops as seen in Figure 3b. This microstructure formed due to the high cooling rates encountered in the SLM process [38,39]. As the cooling rates are smaller for EPBF, Ti-6Al-4V produced by EPBF develops an $\alpha$ + $\beta$ lamellar structure within columnar grains of varying sizes. The structure is known as a Widmanstatten structure and is shown in Figure 3c. Laser and electron beam AM techniques also often develop unique macrostructure like beam path markings indicating the path a laser or electron beam took during part fabrication. Examples for common AM metals such as maraging steel and AlSi10Mg can be found at [40–46].

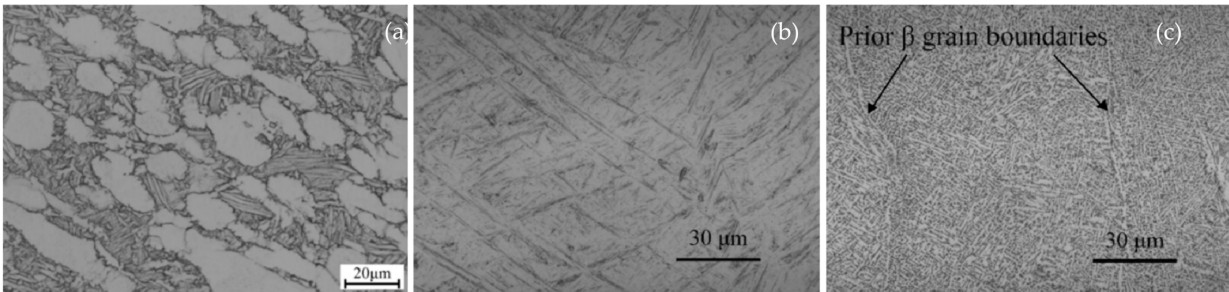

**Figure 3.** Microstructure of Ti-6Al-4V (**a**) from bar stock [37], reprinted from J. Mater. Eng. Perform.; vol. 24, issue 4; X. Shi, W. Zeng, Y. Sun, Y. Han, Y. Zhao, P. Guo; "Microstructure-Tensile Properties Correlation for the Ti-6Al-4V Titanium Alloy;" pp. 1754–1762; (2015); with permission from Springer Nature. Microstructure of Ti-6Al-4V after (**b**) SLM processing and after (**c**) EPBF processing [38], both reprinted from Mater. Des.; vol. 86; H. Gong, K. Rafi, H. Gu, G.D. Janaki Ram, T. Starr, B. Stucker; "Influence of defects on mechanical properties of Ti-6Al-4V components produced by selective laser melting and electron beam melting;" pp. 545–554; (2015); with permission from Elsevier.

Beam-based AM techniques are excellent and proven methods for fabricating small and large metal objects with excellent resolution (fidelity with the digital design). Powder consolidation and laser or electron beam melting coupled with the typically low oxidation of the powders lead to the manufacture of high-density metal parts. The family of beam-based techniques, through extensive research and development, has led to commercialization not only of the techniques but also of the manufacturing modality where companies produce beam-based AM fabricated parts. Recent progress has been made in fabricating building-scale objects using robotic arms equipped with wire and powder feeders and providing the directed energy input. These arms have sufficient degrees of freedom to build large metal objects, i.e., fabrication is not confined to an enclosure typically seen in LPBF and e-beam welding or melting AM techniques. Work continues on different

aspects of beam-based AM including work on multi-material printing [47,48] and large scale AM.

Despite these advancements, beam-based metal printers are too expensive for consumer use unlike FFF polymer printers. Beam-based AM techniques also suffer some technical disadvantages. Defined as layer-on-layer welding processes, beam-based metal AM comes with steep temperature gradients up to $10^6$ K/m and cooling rates in excess of $10^6$ K/s [49,50]. The effects of these gradients are seen in Figure 3c. Due to these high temperature gradients and associated rapid cooling, residual stresses are encountered in the printed part. Although the example shown in Figure 3 was of an AM fabricated material with improved mechanical properties, the cooling rates and temperature gradients encountered were within an acceptable range. If gradients are large enough, part defects could form like distortion or more commonly, cracking within the part. Even if gradients were within an acceptable range, non-optimized parameters such as scan speed or laser energy could lead to balling and/or reduced consolidation (i.e., lack of fusion). Balling leads to rough surfaces due to formation of balls of material caused by high surface tension and inhomogeneous thermal distribution [51–53].

Cracking is another commonly encountered defect, especially when fabricating "non-weldable" alloys with laser powder bed fusion, electron beam melting, or other-directed energy deposition method. In practice, most alloys are weldable (or joinable) if the appropriate welding method is used. For example, oxy-acetylene welding is recommended for cast irons but not for titanium alloys. Whereas electron beam welding is often used for titanium but is not used for cast iron [54]. Due to several factors; however, it is difficult to fabricate objects out of non-weldable alloys. Ni-based superalloys like Astroloy, RR1000, Waspaloy, etc. are susceptible to solidification cracking, liquation cracking, ductility-dip cracking, and strain-age cracking (all-together referred to as "hot cracking" in the literature). The later three cracking mechanisms are typically due to the formation of carbides which act as crack initiation points [55–57]. Solidification cracking is caused along the weld bead and forms when the region surrounding the bead contain both solid and liquid phases. When the region is nearly solidified and is under high tensile loads, the solid phase may restrict liquid from filling in all spaces, i.e., interdendritic zones [55]. For a brief review of the cracking mechanisms and other defects formed (e.g., porosity and balling) by beam-based AM Ni superalloys, see the work by M.M. Attallah et al. [56]. Cracking can also occur in weldable Ni-based superalloys when fabricated by beam-based AM. An example is Hastelloy X which is classified as being weldable with its relatively low Al and Ti content [55,58]. Despite this, hot cracking was observed in AM Hastelloy X [50,59–61]. Work continues on reducing hot cracking in this specific alloy and other Ni-based superalloys when fabricated by a beam-based AM method [62,63]. Hot cracking, porosity, and balling defects have also been observed in other alloys like aluminum AA6061 and AA7075 and stainless steel alloys [64–66]. Liquid metal printing may overcome the limitations of beam-based metal AM especially as it pertains to reduction of temperature related defects such as hot cracking.

### 2.2. History of Inkjet Printing

Inkjet printing is the precursor technology to liquid (or molten) metal 3D printing. The physics behind inkjet printing were developed throughout the 19th century by such imminent scholars as Thomas Young and Pierre-Simon Laplace who discussed the wetting of fluids on a solid surface and the effects of passing liquid through an orifice [67,68]. The Italian physicist, George Bidone, found the shape of droplets emanating from a liquid jet corresponds to the shape of the orifice [69,70]. In 1833, Savart studied the decay in drops and found fluid dynamics governed the breakup of a liquid stream into equal volume droplets by passing it through an orifice [71]. Joseph A.F. Plateau and Heinrich Gustav Magnus studied the effect of droplets passing through a circular orifice [72,73]. Plateau and Lord Rayleigh worked separately on the effects surface area and surface tension has on drop formation [73,74]. Studies performed by these scientist led directly to the

development of the first inkjet printer patented by Rune Elmqvist albeit designed for recording of oscillographs of high frequency phenomenon, such as a heartbeat [75]. From this invention sprang two different methods of jetting printable media, CIJ (schematic shown in Figure 4 below) and DOD printing modes.

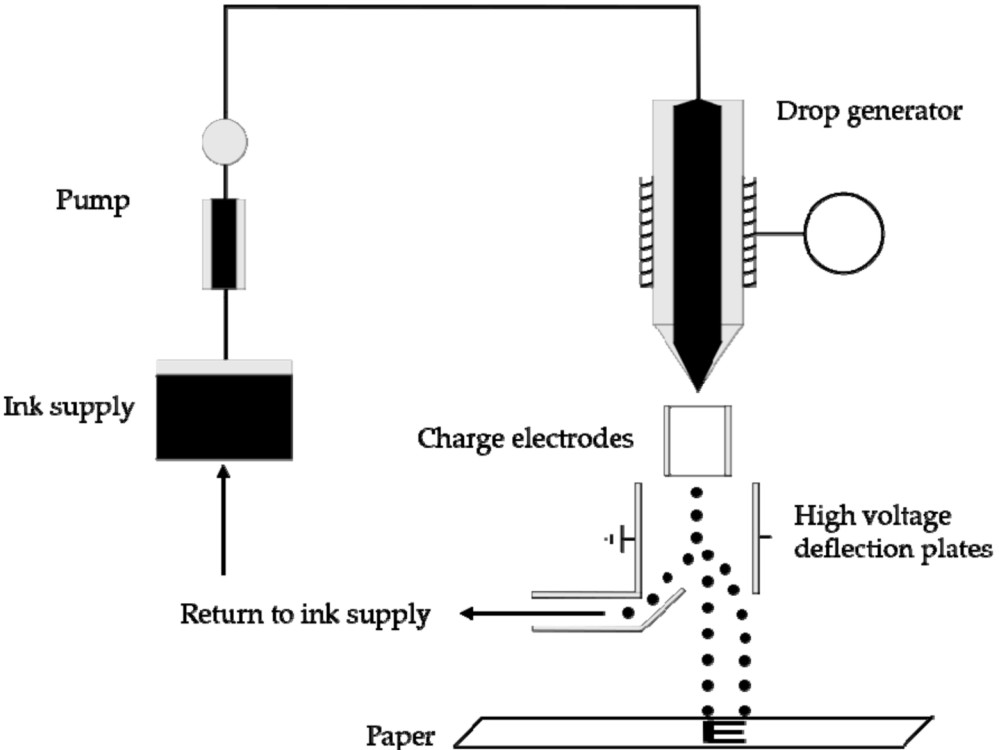

**Figure 4.** Schematic of a continuous jet printer. Ink from the supply is pumped to the drop generator and broken up into drops via an ultrasonic generator. Drops are charged by the charge electrodes and are deflected electrostatically by high-voltage deflection plates.

The first CIJ printers were developed independently by Prof. R. G. Sweet (of Stanford) and Prof. C. H. Hertz (of Lund University in Sweden) [76,77]. These developments lead to the market introduction of the consumer printers IBM 4640 printer and the Iris printer, respectively, during the 1970s [68]. CIJ printers work by first pumping ink from one or more reservoirs into small nozzles forming liquid filaments. Ultrasonic vibrations, usually produced by a piezoelectric (e.g., lead zirconate titanate, PZT) transducer, cause the separation of drops from the filament at regular intervals. This vibration also controls the size of the droplets separated from the liquid stream. An electrically charged plate located near the nozzle charges the drops which are then deflected by application of a voltage to a substrate below the inkjet printer head [78–80]. Stray and excess droplets will go into a gutter and directed back to the ink supply.

Opposite to CIJ of liquid drops is DOD jetting where ink is ejected only when needed. This eliminates the need for a gutter and recirculation system. As there is no need to deflect the ink into the gutter when not actively printing on an underlying surface, there is also no need for charging electrodes and deflection plates (along with the high-voltage source). This means that DOD printers are less complex than continuous jet systems. The first DOD inkjet printers were prototyped by Clarence W. Hansell of the RCA corporation in the 1940s [81]. This DOD device functioned via a pressure wave pushing ink down and out a conical shaped nozzle. The pressure wave is induced by the mechanical deflection of a piezoelectric device consisting of a disc made of a piezoelectric single crystal with electrodes bonded to both sides. This device was never commercialized. The first commercial DOD inkjet printer involved jetting conductive ink by application of a high-voltage pulse through an electrode located outside the printhead nozzle. This method is called the electrostatic

pull inkjet device and was developed and commercialized during the 1960s and 1970s [82]. Due to poor printing performance, companies moved from DOD printers based on the electrostatic pull method to other ink jetting methods, primarily thermal (Figure 5a) and piezoelectric (Figure 5b) driven inkjet nozzles.

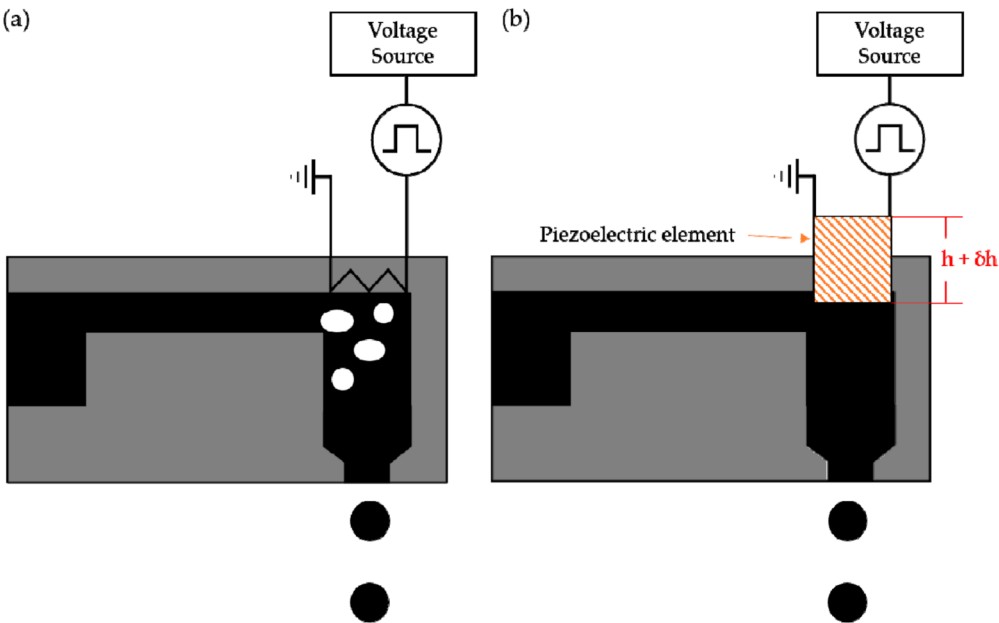

**Figure 5.** Schematic of (**a**) thermal inkjet nozzle and (**b**) a piezoelectric inkjet nozzle. The piezoelectric inkjet is of the push type where polarization of the piezoelectric leads to mechanical deformation and subsequent ejection of ink.

The idea for thermal inkjets was first conceived and developed in the 1960s by Mark Naiman of the Sperry Rand Corporation. Thermal inkjet nozzles jet fluid by passing an electrical current through a wire sitting within the nozzle. The wire heats up to high temperature by resistive heating causing the fluid in contact with the wire to heat up and transform to steam. The steam expands causing a positive pressure in the nozzle pushing ink out the nozzle [83]. The technology was later commercialized by Ichiro Endo of Canon and John Vaught of Hewlett-Packard independently in the late 1970s and early 1980s. John Vaught came up with the idea for thermally jetting ink out of an orifice while observing a similar process in a coffee percolator. At the same time, Mr. Endo and his team at Canon were working on a piezoelectric DOD printing concept. One of the team members accidently touched the tip of an ink syringe with a hot soldering iron, causing ink to jet out due to the high temperature. This accident changed the direction of the Canon team to develop their own thermal inkjet printer. The two companies discovered each other's work and agreed to cooperate on the development of the technology [84,85]. This cooperation, in addition to the simplicity of the thermal inkjet technology, is the reason most consumer printers are thermal inkjet printers.

The other major ink jetting technology is based on pushing or squeezing ink out of a nozzle or inkwell by a piezoelectric element. Applying a voltage to the piezoelectric element causes the piezoelectric to expand against the wall of the inkwell, ejecting ink out of an orifice. This is based on the converse piezoelectric effect where application of an electric charge (really polarization of the element) to a piezoelectric material causes strain in the material leading to mechanical deformation (Figure 5b). The direct piezoelectric effect is the generation of electric charge due to application of force on the material. The two effects are reversible for ferroelectric materials. Most applications requiring piezoelectric actuators involve a ferroelectric ceramic or single-crystal due to the relatively high piezoelectric figure of merit, i.e., piezoelectric coefficient, of these materials. This is true for piezoelectric inkjet printers or droplet generators. Depending on the piezoelectric material, changes to

the direction of the applied polarization will change the direction of the strain. This last point is of note when considering the different nozzle geometries developed for this type of inkjet printing as is discussed next.

Four printhead types, all using a piezoelectric element in difference configurations, were patented in the 1970s and 1980s. The squeeze type piezoelectric DOD inkjet was patented by Steven Zoltran of the Clevite Corporation. In this mode, ink sits within a hollow piezoelectric tube which contracts when polarized, jetting or "squeezing" the ink out of the tube [86]. Separately, N.G.E. Stemme and Edmond Kyser with Stephen Sears of Silonics developed a piezoelectric jet process known as the bend-mode where polarization of the fixed piezoelectric disc or plate causes flexure of the element and a decrease in the volume of the inkwell or chamber. The piezoelectric element sits parallel to the inkwell and "bends" the inkwell jetting ink out of the nozzle [87,88]. A related jetting mode is the push-mode method where the piezoelectric is perpendicular to the inkwell as shown in Figure 5b. This mode of ink jetting was developed by Stuart Howkins of the Exxon Research and Engineering Company [89]. The final piezoelectric inkjet type involves using a piezoelectric element in shear mode. The first three piezoelectric inkjet types polarized the piezoelectric parallel to one of the crystallographic axes and utilized mechanical deformation parallel to either the same axis to that of the polarization or another crystallographic axes. Piezoelectric materials can also be excited in shear mode where polarization occurs as before, but deformation occurs along a shearing direction and is used in the jetting of liquid. Shear type piezoelectric inkjet printheads were patented by Fischbeck and Wright of the Xerox Corporation [90]. For a comprehensive summation of the fluid mechanics, technology, and applications of inkjet printers, please refer to the following reference [91]. Thus far, this background has focused on the jetting of dyes, pigments, or organic solvents used for printing of facsimiles. What follows is a brief background in 3D inkjet printing.

### 2.3. History of Inkjet Type Printing for Metal Additive Manufacturing

Emanuel Sachs et al. first demonstrated printing of solid ceramic parts via an ink-jet process in the early 1990s [21]. In this work, the authors used both continuous and drop-on-demand (DOD) ink-jet systems to print alumina molds for the metal investment casting. Continuous inkjet was found to be preferred for its speed over DOD inkjet printing. The final part achieved over 99% of the final dimensions as compared to the modeled dimensions. The smallest feature size was about 430 μm with the final part size of 5 cm. Later in the 1990s, a group out of the United Kingdom used a piezoelectric-driven DOD printer to print yttria stabilized zirconia parts by jetting ink loaded with 5 to 60 vol% ceramic. The ink was a mixture of zirconia powder, a thermoplastic resin (for improved particle binding during pyrolysis), and a dispersant. The ink was jetted out of 75 μm sapphire nozzles in layers and pyrolyzed at 120 °C. A free-standing, pyrolyzed and sintered zirconia part was printed with 65 layers, each 50–70 μm thick [92]. The same group later demonstrated printing of fully dense titania parts using the same process [93].

The freeform fabrication of a three-dimensional metal object by a material jetting process may have first been demonstrated by Johannes Gottwald of the Teletype Corporation in 1969. In his invention, the "ink" used was made entirely of a Bi-Pb-based metal alloy with a relatively low melting point of 158 °C [94]. This was opposed to the use of a metal loaded dye or pigment. The molten metal was ejected out of a conducting nozzle and through a magnet onto an oxide coated steel substrate. The stream of conductive ink from nozzle to substrate completed a circuit generating a magnetic flux. The ink streams through a magnet connected in series with the circuit generating a second magnetic flux. The two magnetic fields were coupled allowing for deflection of the ink stream enabling generation of a metal object.

Progress in liquid metal freeform printing continued with the jetting of low temperature solder for the electronics industry [95,96]. A patent by Hieber published in 1989 described a method of jetting molten solder via a piezoelectric inkjet system [96].

Marusak developed a piezoelectric actuation scheme for jetting low-temperature Indalloy-58 for micron scale soldering [97]. Work continued through the beginning of the new century on advancing the use of piezoelectric driven actuation methods for molten metal jetting [98–102]. The previous piezoelectric techniques were in modes where the molten metal is directly ejected by expansion of the piezoelectric ceramic or crystal due to application of a voltage across the piezoelectric. Richard C. Oeftering patented an acoustic method of jetting molten metal where a piezoelectric transducer produces acoustic waves which leads to droplet formation and ejection [103]. Due to limitations in the operating temperature of piezoelectric materials; however, insulation is required for higher melting point metal alloys (>300 °C).

Most of the work in molten metal jetting had been on solder deployment for the microelectronics industry [104,105]. Orme and Smith investigated a method of using continuous inkjet technology to jet molten aluminum (Al) to form free-standing structures [106]. This work was based on earlier efforts on printing fully formed, free-standing metal objects via a material jetting method [107–112]. To this author's knowledge, this work is the first-time Al objects (or any other metal/alloy) were printed via a material jetting method to a near-net free-standing form. This work was based on continuous inkjet printing piezoelectric actuated ultrasonic vibration within the fluid to separate individual droplets from the molten Al jet stream. Individual droplets would then be deflected electrostatically. The following sections will review the current state-of-the-art in liquid metal printing and will attempt to focus solely on the jetting of pure liquid metal or metal alloys for the fabrication of free-standing objects. Section 3 will expand on the latest work in CIJ printing of free-standing metal objects. Section 4 will cover piezoelectric DOD-based liquid metal printing. Section 5 details the electrohydrodynamic, magnetohydrodynamic, and electromagnetic DOD methods. Section 6 discusses the many pneumatic-driven DOD techniques. Sections 7 and 8 will cover an impact-driven technique and a family of laser-based metal droplet generators, respectively.

## 3. Continuous Liquid Metal Printing

As indicated in the previous section, Orme and Smith first printed free-standing Al parts using a continuous inkjet printing setup. In their setup, Al is liquified in a cartridge or heating assembly fitted into a droplet generator. The generator sits within a socket assembly located above an environmental chamber. The heating assembly consists of a piezoelectrically driven plunger assembly which is inserted into the metal reservoir. The plunger assembly consists of a vibrating rod with a lead titanate zirconate )Pb(Ti,Zr)O$_3$ (PZT) and, in this case, PZT-5H) piezoelectric ceramic element bonded to the top of the rod. The reservoir has a 100 μm orifice at the bottom of the heater assembly. The whole heater assembly is water cooled and the vibrating rod serves as insulation for the PZT element. A principal concern with molten metals, especially Al, is their highly corrosive nature in the molten state. Orme and Smith found molten Al readily dissolves stainless steel and Inconel while titanium is relatively resistant [106]. Therefore, the cartridge was machined from titanium and for further protection, coated with boron nitride.

The PZT element vibrates when an electric signal is applied. This vibration translates across the whole plunger assembly leading to instability in the liquid stream causing breakup of the stream into metal droplets. The orifice is constructed from diamond which is immersed in an inert argon (Ar) atmosphere. Despite this, after even one hour of exposure to the molten Al, erosion of the diamond orifice was observed in a scanning electron microscope. The temperature at the orifice was maintained at 690 °C (30 °C above the melting point of the Al). Below the orifice is a charge tube which charges the molten droplets like in a standard CIJ printer. Once out of the nozzle, droplets fall approximately half a meter to a copper (Cu) substrate for deposition. The charged droplets are deflected by charged plates in the environmental chamber. The entire environmental chamber where droplets are deposited is filled with an inert atmosphere and maintained at a pressure of 14.7 psi (≈101 kPa or 1.0 atm) [106].

The CIJ method was studied for pure tin (Sn), a zinc-tin alloy [113] and a tin-lead alloy [114]. H.-Y. Kim et al. investigated the splashing/recoiling/bouncing behavior of pure Sn and $Zn_{0.8}Sn_{0.2}$ droplets unto a substrate. The tendency of droplets to recoil or not stick to a surface ultimately affects the size of solidified droplets thus hurting the resolution of a build. They were able to develop a control methodology where the vibration frequency for stream break-up is adjusted to control the droplet size. Their method led to formation of near uniform solid metal drops with a variation of less than 3% in diameter [113]. X.-S. Jiang et al. investigated the relationship between the diameter of micro-droplets and processing parameters of liquid metal alloy jetted with a CIJ printer. They were able to jet a stream of $Sn_{0.63}Pb_{0.37}$ through a 140 μm orifice and gas pressure (applied on the molten alloy) of 30 kPa. If no frequency is imposed on the jet stream, a fiber of the alloy is formed. A frequency of 15 kHz led to droplet formation but no breakup of the stream. The same alloy was jetted through a 100 μm orifice with a gas back pressure of 40 kPa. Droplets with an average diameter of 192 μm were broken from the stream with a frequency of 2.6 kHz [114].

Liquid solder alloys were the first molten metals to be jetted in an inkjet manner with a CIJ printer. In terms of free-standing objects (not wafer or solder bumps), Al was the first molten metal to be printed, again by CIJ. The alternative mode, DOD, has been demonstrated for a greater number of metals; however, as compared to the CIJ mode. The current state of DOD liquid metal printing is discussed in following sections. Several DOD techniques starting with piezoelectric driven jetting technologies followed by electrohydrodynamic, magnetohydrodynamic, electromagnetic, pneumatic, and impact-driven jet printing technologies will be covered. The section on pneumatic DOD printing will include a novel printing method where free-standing metal objects are formed from molten metal droplets cooled in a liquid bath. Lastly, a laser-induced droplet generation technique will be introduced.

## 4. Piezoelectric Drop-on-Demand Printing

For the printing of free-standing metal objects, piezoelectric actuators have been used in a similar way to piezoelectric DOD inkjet printing as discussed in Section 2.2. The work by Richard Marusak on Indalloy-58 involved using a squeeze-mode liquid metal nozzle. In this mode, liquid dispensing through a glass tube with a piezoelectric crystal surrounding the middle of a glass tube. One end the tube was connected to the fluid supply while the other end was the orifice where metal droplets are ejected [97]. The piezoelectric crystal "squeezes" the tube when voltage is applied to it, ejecting droplets with sizes depending on the size of the orifice (~50 μm in the referenced work) and volumes in the hundreds of picoliters, perfect for solder deployment on microcircuits. A schematic is shown in Figure 6a [115,116]. Until recently, piezoelectric squeeze mode printers were capable of printing only metals with melting points below 240 °C.

In an updated design of piezoelectric squeeze-mode printers, Rumschoettel et al. demonstrated the possibility to jet molten metals with melting points up to 600 °C. Their design insulates the piezoelectric from the higher temperatures by placing the piezoelectric element at one end of the glass tube and the molten metal at the other end. The piezoelectric is attached directly to a base plate. Underneath the piezoelectric, the glass tube is attached. A heating element is then placed at the midsection of the tube and is attached to an insulated ceramic heating element to provide the heat to melt the metal in the tube. Jetting takes place by applying an electric signal to the piezoelectric element. The element, typically a piezoelectric ring stack (multiple piezoelectric ring-shaped ceramics "glued" together), expands and contracts along the axial (polar axis) direction. This action generates longitudinal compression waves which are transmitted to the glass tube. The waves travel along the length of the tube to the tip where the waves are transmitted to the liquid. The wave interacts with fluid at the nozzle tip causing a pressure difference leading to droplet formation and ejection [102]. The design allows for higher temperature molten metal jetting, but this comes at a cost, for optimum jetting, the glass tube, fused silica in this

case, must be impedance matched to the molten metal. Therefore, glass tubes must be custom made for each metal increasing the cost of these printers. The gained advantages of higher operating temperatures coupled with the higher jetting frequencies obtained with piezoelectric DOD printers may outweigh this cost.

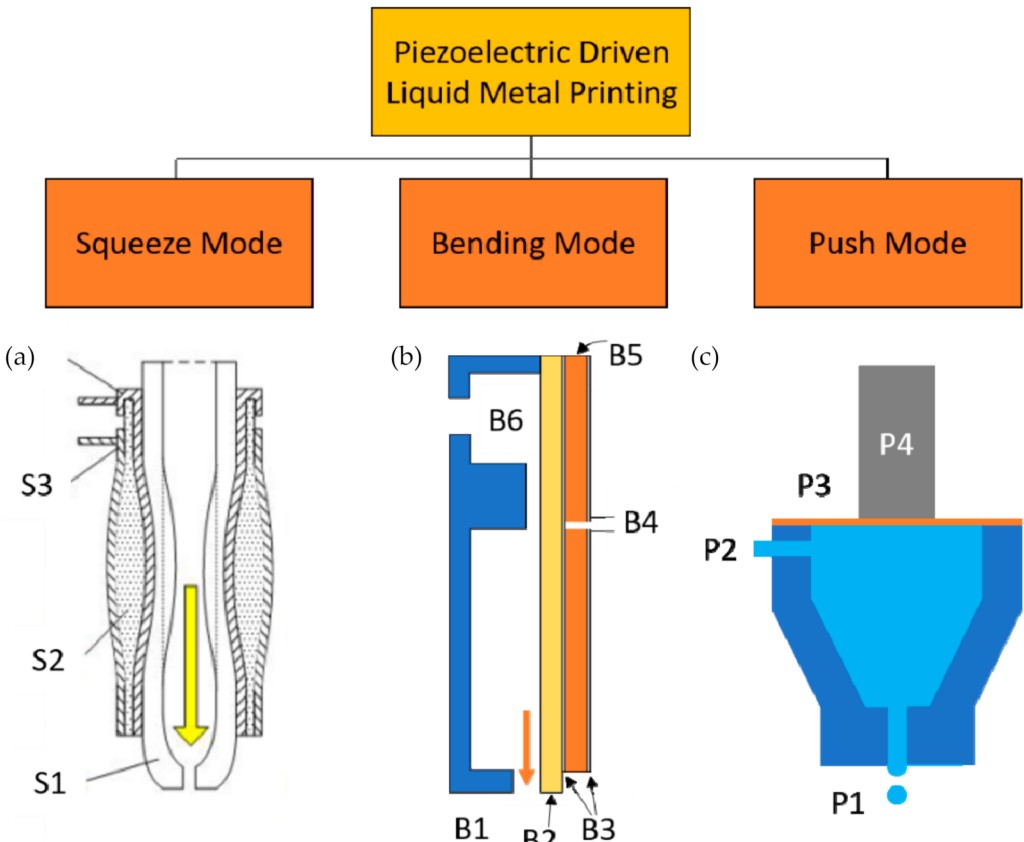

**Figure 6.** Schematic of three different piezoelectric inkjet printhead modes. (**a**) Squeeze mode piezoelectric actuation (S1: Glass tube, S2: Piezoelectric material, S3: Outer electrode, and S4: Inner electrode) [116]; reprinted from J. Manuf. Syst.; vol. 47; T. Wang, Tsz-Ho Kwok, C. Zhou, and S. Vader; "In-situ droplet inspection and closed-loop control system using machine learning for liquid metal jet printing;" pp. 83–92; (2018); with permission from Elsevier. (**b**) Bending mode piezoelectric actuation (B1: Support frame, B2: Diaphragm, B3: Electrodes, B4: Voltage leads, B5: Piezoelectric material, and B6: Jetting chambers). (**c**) Push mode piezoelectric actuation using a diaphragm as opposed to a piston (P1: Nozzle, P2: Molten metal inlet from a reservoir, P3: diaphragm, and P4: Piezoelectric material).

Several other piezoelectric printing modes exist including bending mode, shear mode, and push mode. Bending mode piezoelectric actuators (schematic seen in Figure 6b) have been developed for jetting metallic inks and low temperature molten metal. Electrodes are applied to both sides of a piezoelectric. The electroded piezoelectric is attached to a flexible substrate like a metal diaphragm (called a bimorph). Sitting below the diaphragm the jetting chambers which consists of a throttle, a pumping chamber, and the nozzle. These chambers are connected to the reservoir via a fluid inlet port. When voltage is applied to the piezoelectric, the bimorph bends due to mechanical deflection of the piezoelectric. The primary applications for this bimorph (piezoelectric film or thin plate attached to a substrate which amplifies bending) bending actuator includes microelectromechanical systems (MEMS), microlens arrays, and low-temperature solder deposition [117–121]. Shear mode piezoelectric actuators have been developed for inkjet applications [122]; however, to this author's knowledge shear mode printheads have not been used for the jetting of molten

metal. This might be due to the difficulty in fabricating these types of actuators coupled with the relative ease in jetting fluid with a push or squeeze mode actuator.

More commonly used piezoelectric actuators for molten metal jetting is in a push-mode format as seen in Figure 6c. Jetting metal droplets with a piezoelectric push-mode printer was proposed by Yamaguchi et al. who printed hemispherical drops of a fusible alloy consisting of Bi, Pb, Sn, Cd, and In [123]. Yokoyama et al. developed a molten metal ejector for soldering in electronic packaging applications [98]. Their device works by ejecting molten solder (Sn-3Ag-0.5Cu) from a heated chamber. At the top end of the chamber is a diaphragm attached to a piezoelectric ceramic element. When a voltage is applied to the element, it displaces the diaphragm leading to liquid ejection. The authors stated their element had a Curie temperature of 345 °C which likely means the piezoelectric is made of a composition of lead zirconate titanate (PZT), a ferroelectric material. A drawback to ferroelectrics is the relatively low temperature phase transition (associated with the Curie temperature) between the ferroelectric and paraelectric phases, which for PZT (depending on the composition) is at most 380 °C. The maximum operating temperature for PZT and similar ferroelectrics; however, is typically around $\frac{1}{2}$ the Curie temperature or transition temperature. Therefore, the maximum temperature for molten metal allowed in these setups would be at most 190 °C. Lee et al. showed how to work around the temperature limits of a piezoelectric by insulating the piezoelectric and attaching a piston which is in contact with the molten metal as opposed to the diaphragm and piezoelectric [100,101]. Their piezoelectric DOD printer was also a push-mode printer with a piston insulating the piezoelectric from the molten metal. They rated their printer's maximum operating temperature at 400 °C and printed high aspect ratio columns of 63/37 Sn/Pb solder with heights exceeding 4 mm.

Piezoelectric DOD printers can print many liquid solders including both Pb-based and Pb-free solders. The solder alloy, Indalloy-158, was the first metal alloy to be printed via an inkjet DOD process. The piezoelectric squeeze-mode DOD printer used to print solder bumps of Indalloy-158 had a reservoir lined with polytetrafluoroethylene (PTFE, $T_m$ = 327 °C), also known as Teflon™. The PTFE lining set the maximum temperature for the printer [97]. A push-mode piezoelectric system was shown to jet droplets with sizes as low as 70% of the nozzle diameter. Yokoyama et al. demonstrated control of droplet volume through adjustment of the drive signal to the piezoelectric element. This was shown to generate droplets with average diameters of 35 and 68 μm for nozzles with 50 and 100 μm diameters, respectively [98].

## 5. Field Induced Drop-on-Demand Printing

### 5.1. Electrohydrodynamic Drop-on-Demand Printing

In this section, three DOD jetting methods based on application of a field to jet molten metal will be discussed. The first, electrohydrodynamic (EHD) printing, involves the use of applied electric fields in the jetting of inks or molten metals. The method works by applying an electric field to a conductive fluid passing through a small opening or orifice. If the electric field is strong enough to overcome surface tension in the fluid, the fluid itself will take the shape of a cone outside the orifice. The shape is known as a Taylor Cone [124]. A schematic of the formation of a Taylor cone is shown in Figure 7 where initially, a small charge causes the liquid to form a meniscus or pendant shape. This shape is exaggerated as the charge increases and when the charge is sufficiently high, a Taylor cone is formed. Liquid jetting occurs at the end of the cone at voltages near a threshold voltage for the formation of the Taylor cone [125].

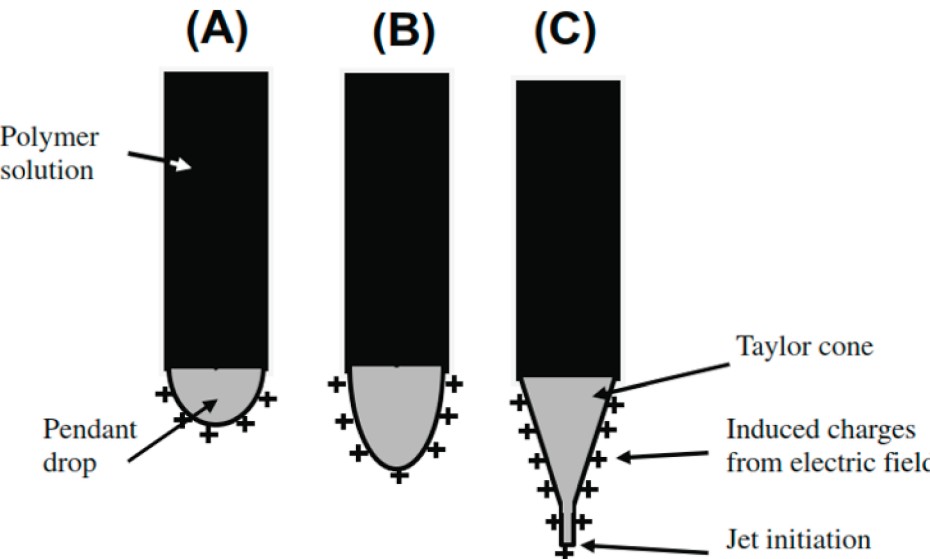

**Figure 7.** Schematic of the formation of a Taylor cone in a polymer solution [125]. In (**A**), an electric field is applied causing charges to migrate to the surface of the solution. The surface elongates due to the charges in (**B**) and with an increase in the applied field deforms into a cone shape, the Taylor cone as seen in (**C**). Reprinted from Compos. Sci. Technol.; vol. 70, issue 5; A. Baji, Yiu-Wing Mai, Shing-Chung Wong, M. Abtahi, P. Chen; "Electrospinning of polymer nanofibers: Effects on oriented morphology, structures and tensile properties;" pp. 703–718; (2010); with permission from Elsevier.

For high-resolution printing via the EHD method, voltages near the threshold value are ideal. When the voltage increases past the threshold, additional droplet formation occurs. The use of Taylor cones for the fine spraying or ejecting of liquid streams are seen in several applications including electrospinning, electro-spraying applications, and focused ion beams (which for many FIB systems uses liquid gallium as a source of ions for micron-sized milling of samples). For AM, the big advantage of using EHD is the very high resolutions which could be achieved as the liquid stream emitted from the tip of the cone could be orders of magnitude smaller in size than the nozzle.

To date, most of the work on EHD printing involves printing of polymeric materials and/or biomaterials or the jetting of metal nanoparticles in a way similar to the jetting of conductive inks in inkjet printers [126–131]. Han and Dong; however, developed an EHD method of jetting molten metals. They showed a printer capable of jetting a low temperature Pb-free solder although they claimed their system could handle temperatures as high as 385 °C [127]. A schematic of their setup is shown in Figure 8a [127,132]. In this work, the authors demonstrated the jetting of Field's metal, which is a ternary eutectic alloy of 51 wt% In, 32.5 wt% Bi, and 16.5 wt% Sn. The molten metal (T = 62 °C [133]) is placed in a syringe. The syringe is connected to an air pressure regulator on one end, which maintains pressure on the molten metal. A heating coil is wrapped around the syringe and connected to a temperature controller to maintain temperatures adequate to keep the metal in a liquid state. At the other end of the syringe is the nozzle with a connection to ground. A voltage is applied to the nozzle which generates an electric field, in turn creating a Taylor cone. Molten metal droplets are formed, deposited, and cooled on a movable substrate attached to a movable X-Y positioning stage. Their work resulted in the fabrication of several metal objects including a free-standing metal column (Figure 8b), a thin wall of metal (Figure 8c), a metal bridge structure with thicknesses near 100 μm (Figure 8d), and metal wire laid out in an interdigitated pattern (Figure 8e) [132].

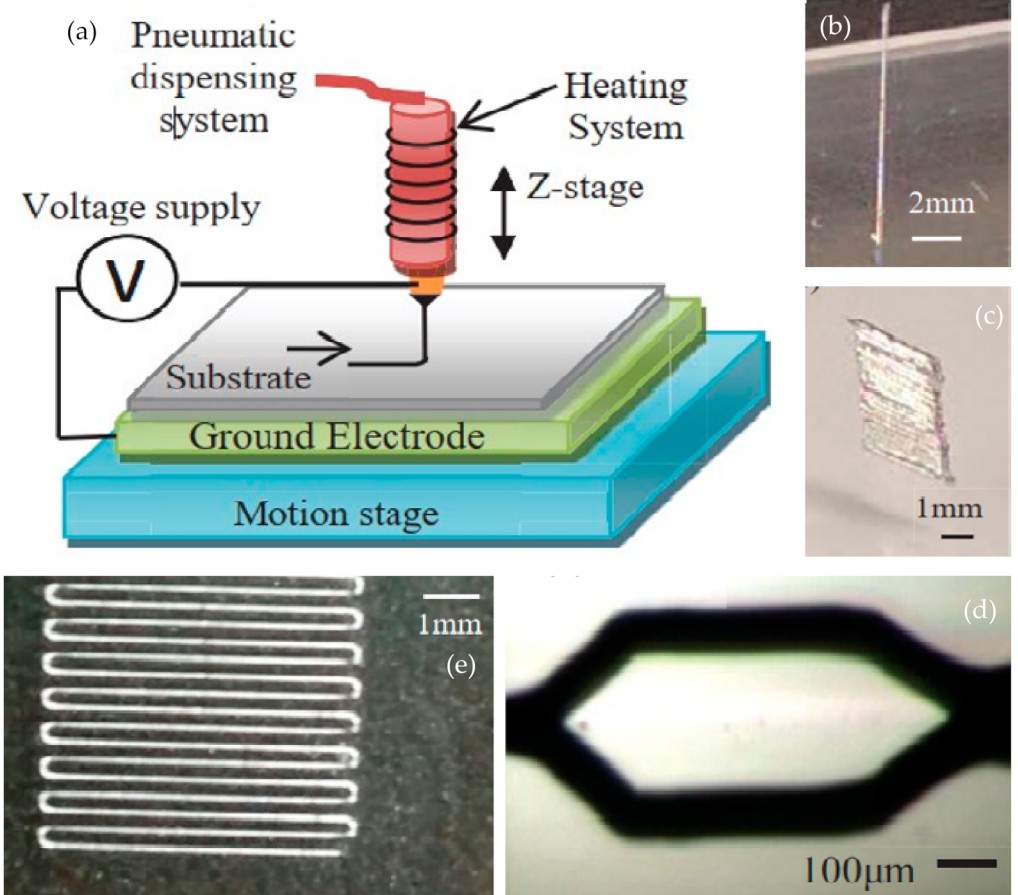

**Figure 8.** Diagram of the system (**a**) for the direct printing of metal through electrohydrodynamic jetting of molten metal. The system printed metal objects including: (**b**) a pillar, (**c**) a wall, (**d**) a metal bridge, and (**e**) wire printed in a complex pattern [127]. Reprinted from Procedia Manufacturing, vol. 10, Y. Han and J. Dong, "High-Resolution Electrohydrodynamic (EHD) Direct Printing of Molten Metal," pp. 845–850; (2017); with permission from Elsevier.

### 5.2. Magnetohydrodynamic Drop-on-Demand Printing

The second method of applying a field to jet molten metal is a relatively recent development in beamless, non-contact, metal freeform fabrication. It is a printing method based on the magnetohydrodynamic jetting of molten metal. The general method utilizes a magnetic field induced pressure gradient to jet liquid metal droplets. Two printer systems have been developed using a magnetic field in this way for the jetting of liquid metal drops, MetalJet developed by Rasa et al. of Oce-Technologies BV of the Netherlands [10,134] and MagnetoJet™ commercialized by Vader Systems [135], which was recently acquired by Xerox [136].

The first of the two technologies developed was the MetalJet system devised by Oce-Technologies BV. A schematic of their system is shown in Figure 9. Liquid metal is melted in a cartridge through induction heating indicated by the wire wrapped around the cartridge chamber. The liquid metal is then ejected from the ejection chamber of the nozzle by an induced Lorentz force. The ejection chamber is surrounded by permanent magnets inducing a magnetic field within the fluid. Perpendicular current density and Lorentz force vectors are generated. The Lorentz force places pressure on liquid in the nozzle tip creating a fluid stream. The stream is broken up by pulsing an electric current provided by tungsten (W) electrodes located at the tip of the nozzle. The W electrodes are in physical contact with the liquid metal. The pulsing current is controlled by a waveform generator and a current amplifier. Ejected droplets impact a substrate below the nozzle attached to

a motorized stage. Parts are fabricated in a layer-wise fashion following a digital pattern without the need for a support structure as used in LPBF.

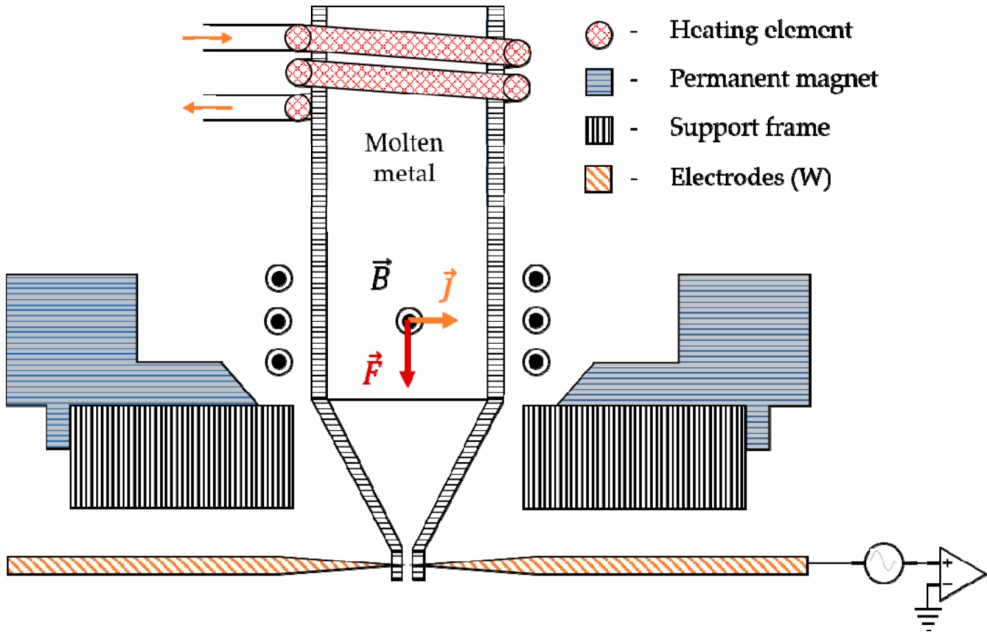

**Figure 9.** Schematic of a magnetohydrodynamic DOD MetalJet printer. Metal feedstock is melted through induction heating and ejected out the nozzle by a magnetic field induced force. Schematic based on patent for the MetalJet nozzle [134].

The MetalJet system is capable of printing millimeter sized objects with a resolution of about 80 μm. Simonelli et al. also showed their system is capable of printing liquid metal with melting points as low as Sn up to ≈1000 °C (shown on silver, Ag) [10]. In this research, the authors found that the choice of substrate will dictate whether the droplets will require post machining or not. They used a Cu substrate and found Sn droplets will adhere while Ag droplets will not adhere to the substrate. More importantly, droplet to droplet fusion behavior varied between Sn and Ag. Significant fusion between droplets was observed in Sn while little fusion was observed for Ag. The surface roughness varied with an order of magnitude increase in roughness observed for Ag droplets as compared to Sn. For objects fabricated with both metals, sintering was required for densification of printed objects.

A related system, shown in Figure 10, designed by Vader Systems (now a part of Xerox) used a magnetic field to generate pressure on molten metal leading to droplet formation and jetting out of the nozzle. A computer controller with the digital design controls the position of a heated substrate to which hot molten droplets are deposited. Simultaneously, the controller feeds metal wire from a spool into a ceramic reservoir at a predefined rate. The ceramic reservoir is resistively heated to melt the metal, which is then fed into the nozzle through capillary forces. An electromagnet in the form of Cu wires is coiled around the nozzle. Pulses of electric current is applied to the Cu coil generating a transient magnetic field. The field induces and couples with a circulating current density generating a Lorentz force leading to a pressure pulse which ejects droplets of the molten metal. The reservoir and nozzle are enclosed in an Ar gas shroud. The gas shroud prevents oxidation and provides directional stability in the stream of metal droplets ejected from the orifice of the nozzle [135,137–140]. A schematic of MagnetoJet™ is shown in Figure 10a. Free-standing objects, a rectangular bar (Figure 10b) and a twisted helix cup (Figure 10c), were both printed of Al4043 with this system [139].

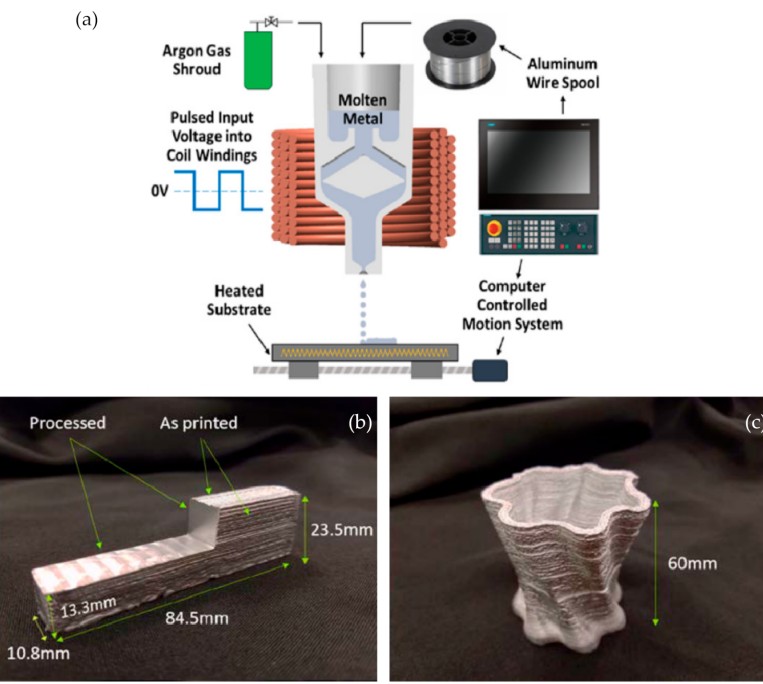

**Figure 10.** (**a**) Schematic of a magnetohydrodynamic DOD printer. Metal wire is fed into a heated chamber at a controlled rate. A magnetic pulse leads to ejection of molten metal droplets down to a heated substrate. Printed objects include (**b**) a multi-level rectangular bar and (**c**) a twisted helix cup [139]. Reprinted from "Magnetohydrodynamic Drop-on Demand Liquid Metal 3D Printing," by V. Sukhotskiy, I. H. Karampelas, G. Garg, A. Verma, M. Tong, S. Vader, Z. Vader, and E. P. Furlani (2017).

The nozzle used in the MagnetoJet™ system was made of a refractory material with a lower orifice diameter between 100 and 500 μm. The droplet size ranged from 50 to 550 μm. The size and shape of metal droplets generated in both MHD systems depended on the interior geometry of the nozzle, the shape and size of the nozzle orifice, ejection frequency and pulse duration [10,140]. The differences between the two systems include location where droplets split from the jetting stream and what occurs at the substrate. The MetalJet system used W electrodes just below the bottom of the nozzle orifice. The liquid metal stream was then broken up into droplets by applying current pulses to the electrodes. In the MagnetoJet™ setup, ejection coils surround the nozzle providing the Lorentz force to push liquid out the nozzle and control the formation of the droplets. The results of droplet size and resolution appear to be similar across both systems. The key difference is in the substrate. As noted for the MetalJet system, choice of substrate is crucial to realize higher density parts. The substrate in the MagnetoJet™ system was heated and this provided a thermal bath to control droplet spread, cooling rate, and fusion. It remains to be seen whether heating the substrate will help in droplet fusion of metals such as Cu or Ag with higher melting points as compared to Al.

### 5.3. Electromagnetic DOD Printing

The first two field induced DOD printing methods used one field to induce jetting of liquid metal. EHD actively jetted droplets using an applied electric field while MHD uses an induced magnetic field to jet droplets. In the third field driven DOD method, developed by Luo et al., both an electric field and a magnetic field are applied to induce and jet molten metal droplets [141]. In electromagnetic DOD printing, liquid metal is externally supplied to the chamber. In this case, mercury (Hg) was tested as was an unidentified metal solder, which was liquid at 290 °C. Inserted into the chamber on opposite ends are two electrodes providing the electric current (up to 40 A from a 12 V source) while permanent magnets are placed on the sides of the chamber to provide the magnetic field with an intensity of

0.4 T. The authors used a fixed pulse duration of 5.0 ms and found the droplet diameter (≈350 μm) was constant at printing frequencies between 0 and 150 Hz. Droplet diameter did change with nozzle diameter, pulse width, and magnetic field intensity. Droplet diameters as low as 50 μm were achieved when nozzle diameter was 100 μm.

## 6. Pneumatic Drop-on-Demand Printing

### 6.1. Low Temperature Pneumatic Techniques

Previous droplet generators and liquid metal printers require application of electric fields for the separation and/or generation of metal droplets. EHD required an electric field for the creation of Taylor cones while in MHD printers, an electric current through metal wires generated magnetic fields used to create a Lorentz force on molten metal leading to droplet generation. This requirement for electricity adds complexity to the design of the printer. For piezoelectric actuated DOD printers, a piezoelectric element is needed requiring a power supply and depending on the metal jetted, requires insulation. CIJ printers would also need power supplied to the nozzle and deflection plates to electrostatically separate drops from the stream and deflect droplets following a digital pattern. A simpler method would be to apply adequate pressure on one end of the molten metal forcing liquid out a nozzle. Issues revolving around fluid viscosity, nozzle size, oxidation, temperature, etc. would need to be addressed for each proposed metal; however, in principle, many metal alloys could be printed by a simple pneumatic method.

A research group out of North Carolina State University studied two methods of jetting near room temperature liquid metal, EGaIn, a binary eutectic alloy with a melting point of approximately 15.7 °C [142]. The first method, "syringe method", used a syringe pump and piston to dispense the liquid alloy out a syringe needle (26s or 33 gauge). The syringe first imbibes liquid alloy. A drop of alloy is deposited on the substrate to form a footprint and the syringe is withdrawn from the substrate forming metal wires. To maintain downward pressure, careful coordination between the syringe assembly and the stage in which the substrate sits must be maintained. The second method ("pressure method") uses gas pressure to dispense liquid alloy out of a glass micro-pipette epoxied to a syringe needle (27 gauge). In this method, shown in Figure 11, liquid alloy is again placed in a syringe. Hydrostatic pressure is controlled and changed by changing the height of fluid in the syringe. Gas pressure above the liquid is held constant except to eject an initial droplet (small pressure pulses are applied) to serve as a footprint for the subsequent metal wire.

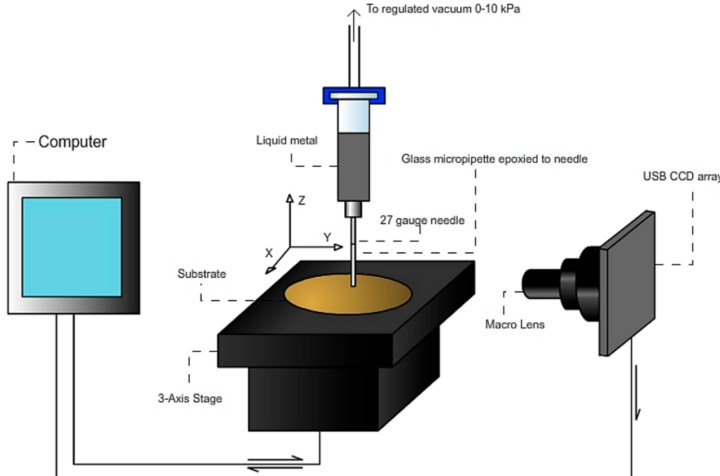

**Figure 11.** Schematic of the pressure method used in jetting the binary eutectic alloy EGaIn. The liquid metal is placed in the syringe and gas pressure is applied to the "head" of the syringe above the liquid [142]. Reprinted from supporting information (Figure S2) of Adv. Mater.; vol. 25; C. Ladd, Ju-Hee So, J. Muth, and M. D. Dickey; "3D Printing of Free Standing Liquid Metal Microstructures;" pp. 5081–5085; (2013); with permission from John Wiley and Sons.

Both methods were shown to fabricate free-standing EGaIn metal wire with lengths approaching 1.0 cm in height. Both methods began by ejecting a single footprint droplet and then building metal wire up from the initial drop. The pressure method was easier to operate as opposed to the syringe method. The reason for this was that in the pressure method, the height of the fluid in the syringe controlled the hydrostatic pressure so liquid jetting was independent of the motion of the stage so long as the forming metal wire was in tension. In the syringe method, a piston was used to eject liquid requiring the more precise coordination between stage and ejector. More intricate structures were also fabricated following several different jetting approaches including rapid expulsion of metal or direct stacking of metal droplets to form metal objects. Another method, reminiscent of photolithography, involves first injecting liquid metal into a cast or "microchannel," made of material which is later etched or chemically removed leaving a free-standing metal object. The pneumatic method follows earlier work on ejecting metallic inks through a syringe using air pressure [143].

Another jetting technique for near room temperature metal alloys was recently developed by W. Lei and L. Jing [144]. Like in the previous work (Figure 11), a syringe was used to jet a near room temperature liquid alloy, in this case, $Bi_{35}In_{48.6}Sn_{15.9}Zn_{0.4}$. Additionally, as before, constant gas pressure is applied to the liquid in the syringe. The gas used was N2 supplied from a cylinder and controlled by a solenoid valve. As the melting point of this indium–bismuth quaternary alloy is $\approx 60\,°C$, clogging of the syringe would occur. To resolve this, the syringe was placed in an Al cylinder and a metal coil was wrapped around the outside of the cylinder. The metal coil was connected to a temperature controller feeding electrical power to the coil (made of Cu) for resistive heating of the alloy within. The temperature of the liquid alloy was maintained between 80 and 90 °C [145]. The novelty of this pneumatic method was the quick solidification of the droplets. The end of the syringe where molten metal droplets eject sits in either water or ethanol. The cooling liquid is at room temperature, so liquid metal droplets are cooled upon ejection from the syringe. This allows for rapid cooling of the metal droplets and shorter fabrication times as compared to other DOD techniques.

### 6.2. High Temperature Pneumatic Techniques

For metals with temperatures higher than 300 °C, piezoelectric actuation becomes difficult and without proper insulation, impossible. This is because PZT, often used in piezoelectric actuation applications, exhibits a solid phase transition between 150 °C and 300 °C, depending on composition. This transition temperature, called the Curie transition temperature, causes PZT and similar materials to transition from a non-centrosymmetric (tetragonal) phase to a cubic phase which is paraelectric, i.e., non-piezoelectric [146]. Additional issues include mechanical stresses induced by repeated heating and cooling cycles. Increased susceptibility of oxidation of metal is another concern in the jetting of molten metals. The materials used in the EHT liquid metal droplet generator restricts its use with higher melting point metals as with the piezoelectric actuated printers and droplet generators.

A pneumatic actuation technology, called StarJet, developed by Metz et al. resolved these issues by jetting molten metal out a star-shaped nozzle using an inert gas [147,148]. Figure 12 shows a diagram of the StarJet system and an SEM micrograph of the bottom of the star-shaped nozzle where molten metal is ejected [149]. Deep reactive ion etching is used in the fabrication of the nozzle chip. Placed above the nozzle chip is the liquid reservoir. Etched into the nozzle chip leading to the actual nozzle is a series of channels (gas supply channels) which are connected to side grooves (labeled in Figure 12b). Two gas inlets pump gas into the printhead. The actuation inlet pumps gas both into the reservoir and the rinse inlet pumps gas into the gas supply channels. Liquid is held in the reservoir by capillary forces. When a gas is pumped in the actuation inlet, it pushes down on the liquid. At the same time, gas from the rinse inlet flows into the gas supply channels.

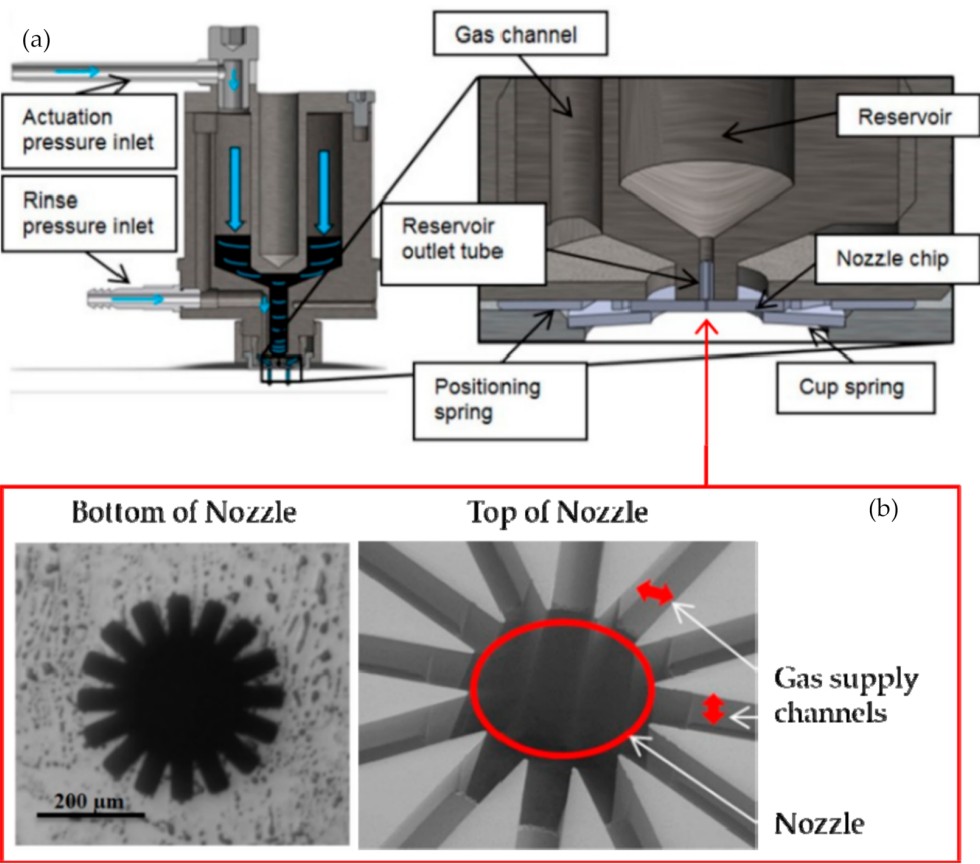

**Figure 12.** (**a**) Diagram of the StarJet printhead and (**b**) an SEM micrograph of the nozzle where molten metal is jetted out [149]. Adapted from Micromachines; vol. 4; "Enhanced Liquid Metal Micro Droplet Generation by Pneumatic Actuation Based on the StarJet Method;" by N. Lass, L. Riegger, R. Zengerle, and P. Koltay; pp. 49–66; (2018); reprinted under CC BY-SA 3.0.

A limiting factor in the various inkjet systems discussed so far is temperature. The transition temperature of piezoelectrics restricts the materials which can be printed using piezoelectric-based DOD printers. No such limitation is known to exist with EHD printers; however, to date metals with melting points below 400 °C have been printed using EHT systems. The printer parts used in piezoelectric and EHT systems restrict the maximum temperature of the molten metal. This is true for any printing technology. For pneumatic driven printers; however, metal alloys with higher melting temperatures can be printed as the printer parts in direct contact with the molten metal, such as silicon ($T_m$ = 1415 °C) in the Starjet system [148], typically have higher melting points. Despite the relatively high melting point of silicon (Si) and because Si is not the only material in contact with the molten metal, important metal alloys such as steel are beyond the ability of the Starjet system to print [148–150].

The highest temperatures reached with the Starjet system was high enough to melt Sn and Al. For metal alloys with higher melting points like Cu require more refractory materials for the crucible and nozzle. Songyi Zhong et al. developed a pneumatic drop-on-demand device for the droplet generation of Cu droplets [151,152]. In their design, Cu is melted by induction heating in a graphite crucible surrounded by coils. Gas is supplied to the crucible via a solenoid valve connected to a T-junction with a vent valve on the opposite side to the solenoid valve. Gas entering the crucible provides the pressure to eject fluid out of a graphite nozzle. The pressure is induced as pulses into the crucible and controlled by a function generator which controls electric pulses supplied to the solenoid valve. This work focused on how processing parameters, supply pressure and pulse width, affected droplet formation and velocity out of the nozzle. They found that droplets form only when the

supply pressure is higher than a threshold pressure. The velocity also increased with an increase in pressure. Increases to the electric pulse width, controlled through the function generator, increased time pressure in the crucible remained above threshold. It was also found that if these two parameters were set too high, multiple droplets would form instead of a single droplet at the nozzle hurting print resolution.

For liquid metal printing of steels, a similar system based on pneumatic actuation of molten metal droplets was developed. Seen in Figure 13a is a schematic of a high-temperature droplet generator developed for AISI 52,100 steel droplets and based on a patent submitted by S. Chandra and R. Jivraj [153]. Like in the previous design for Cu droplet formation, a crucible holds the molten steel. In this design, an alumina crucible is used and was surrounded by a graphite susceptor. The susceptor is surrounded by induction coils which heats up the susceptor to high temperatures through inductive heating. In turn, the heated susceptor heats up the crucible conductively. Heating the metal to be printed in this way allows low melt volumes to remain liquid at elevated temperatures and leads to stable droplet generation. The top of the crucible is connected to a solenoid valve which is connected to a source of inert gas, $N_2$ in this study. The nozzle is found at the bottom of the crucible. The whole apparatus was placed in a vacuum chamber filled with an inert atmosphere [154].

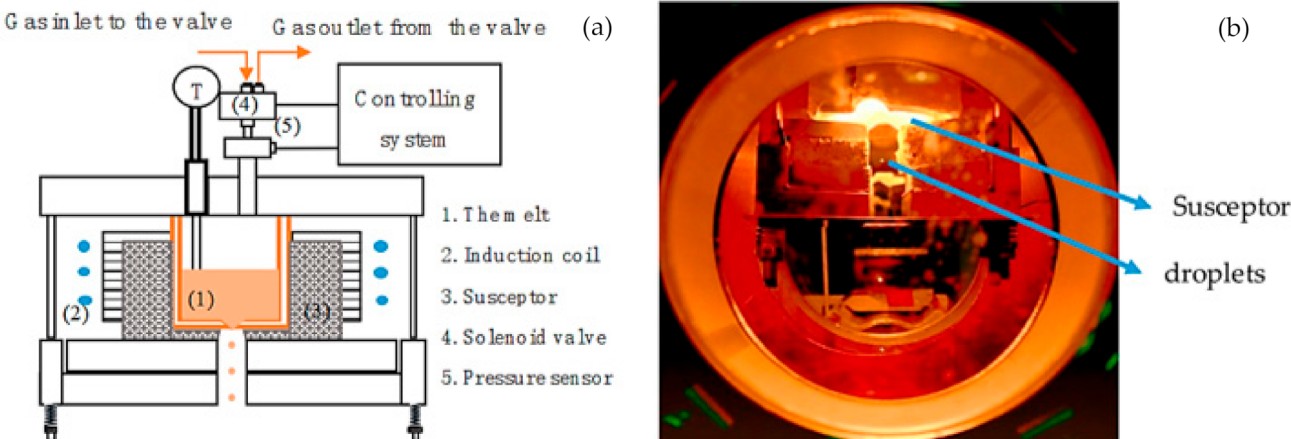

**Figure 13.** (**a**) Schematic of high-temperature DOD printer. (**b**) A view of the inside of the droplet generator with molten steel droplets visible [154]. Reprinted from Micromachines; vol. 10, issue 17; "A High Temperature Drop-On-Demand Droplet Generator for Metallic Melts;" by S.I. Moqadam, L. Mädler, and N. Ellendt; pp. 477; (2019); under CC BY-SA 4.0.

The solenoid valve would be opened for times between 1 and 6 ms and then closed. Cold inert gas is let in the crucible, while the valve is open, and begins to heat up leading to pressure increases due to gas expansion. Once the valve is closed the pressure drops and depending on the melt temperature, drops below an ambient pressure level. The authors referred to this negative pressure as "under pressure." As a result of the pressure changes, a liquid ligament forms below the nozzle. For the high temperatures used for 52,100 steel (≈1560 °C), no under-pressure was observed, and droplets separated from the ligament due to their own inertia. The size and shape of the steel droplets depended on the open time of the solenoid valve as this time affects the dynamic pressure in the crucible. On average, cooled solidified droplets were more spherical with longer valve opening times (i.e., higher pressures); however, particle diameters were smaller on average with longer opening times [154].

## 7. Impact-Driven Drop-on-Demand Printing

A novel DOD technique was recently developed by Luo et.al. where molten metal is jetted by the force generated by a vibrating rod within the fluid above the nozzle [155]. The system, shown in Figure 14, is based on transferring elastic waves to the fluid for

ejection of droplets like the piezoelectric DOD systems. Instead of a piezoelectric element either directly or indirectly (through a connected piston) ejecting fluid, here an impactor is used. This impact DOD system consists of three major parts: an impactor rod, a vibration transfer rod and piston, and the crucible are seen in Figure 14a. The impactor is water cooled and driven by a solenoid. A pulsed voltage applied to the solenoid generates an electromagnetic pulse driving the impactor. A pulse generator supplies the voltage to the solenoid. The impactor hits the top of vibration transfer rod. The top of the rod is shaped in a ball and sits on top of springs (Figure 14b) or a metal ring (Figure 14c). In essence, the transfer rod is a mass-spring system designed like a damped harmonic oscillator to transfer mechanical waves down to the piston. The piston sits at the bottom of the transfer rod just above the top of the nozzle and is in direct contact with the molten metal. Most of the transfer rod and the piston sits within the crucible. The temperature within the crucible is controlled by a heater and a thermocouple. The maximum temperature was tested up to 1227 °C, high enough to melt Ag and Cu. The whole system is shrouded in Ar gas to prevent the liquid from oxidation.

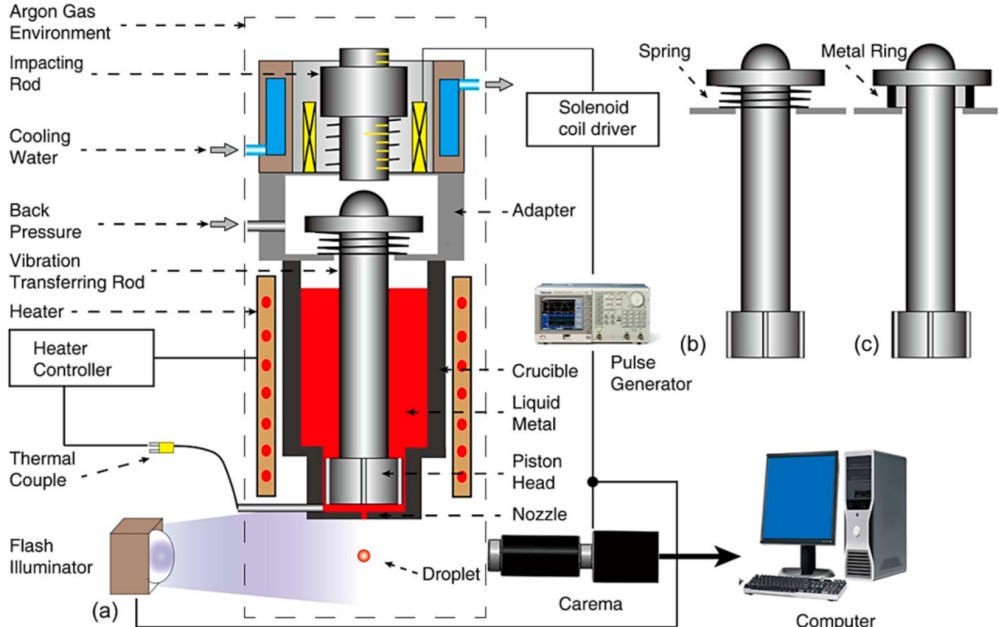

**Figure 14.** Diagram of the impact driven DOD droplet generator. (**a**) A schematic of the whole system including a flash illuminator and camera, pulse generator, and control computer. The top of the transfer rod was fitted with either (**b**) springs or (**c**) a metal disc [155]. Reprinted from International Journal of Machine Tools and Manufacture, vol. 106, J. Luo, L. Qi, Y. Tao, Q. Ma, and C. W. Visser, "Impact-driven ejection of micro metal droplets on-demand", pp. 67–74; (2016); with permission from Elsevier.

The authors jetted a Sn-Pb alloy (60 wt% Sn, 40 wt% Pb) with a melting point around 190 °C. The liquid was held at 400 °C for the conducted experiments. Numerical studies coupled with laboratory experiments showed droplet shape and number was controlled by the pulse width and the amplitude of impact from the impactor. The authors plotted these parameters and identified ranges of pulse width and amplitude where single tear shaped droplets were jetted. Shorter pulse widths required larger amplitudes to generate a single droplet. If the amplitude was too high, ejection of multiple droplets occurred. For longer pulse widths, smaller amplitudes led to single and multiple droplet ejection. It was also observed that to obtain droplet diameters smaller than the nozzle diameter, narrow pulse widths were needed. They were able to achieve droplet diameters around 55 μm ejected from a 100 μm nozzle [155].

## 8. Laser-Induced Droplet Generation

The CIJ and DOD liquid metal printing methods discussed so far jetted metal alloys which were in a liquid state pre-jetting. The following droplet generation method is different in that the metal to be jetted is a solid beforehand. A focused laser is used to liquify metal films forming droplets. The use of a laser to generate molten metal droplets has been explored for several decades. In 1986, Bohandy et al. first reported the direct deposition of metal droplets onto a substrate by laser induced localized melting of a donor film [156]. Their work was motivated by issues surrounding laser assisted deposition of metal films using organometallic gas or liquid precursors. Often such precursors are highly toxic and for certain metals, e.g., gold (Au), Ag, and Cu, available precursors are limited. To resolve these issues, a technique was developed where pulses of a high energy laser melts small regions of a metal film to generate droplets which would deposit directly onto an underlying substrate. This droplet generation technique is now known as "laser assisted forward transfer" (LIFT) coined by the same group [157]. Figure 15 represents a general schematic of the LIFT process. A metal film is deposited onto a transparent support like a quartz slide. The combination of support and metal film is termed the donor. The LIFT technique first used an excimer laser with a quartz cylindrical lens to focus a beam onto a donor. The focused beam would ablatively eject metal from the donor onto a substrate (sometimes termed receiver) below the metal film.

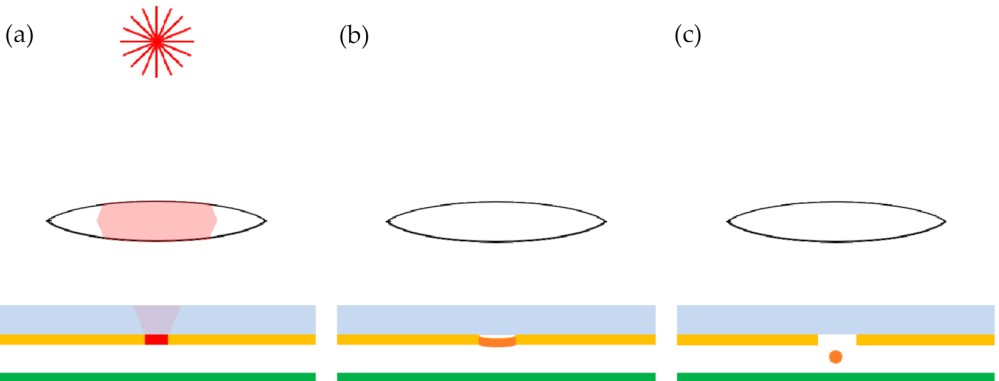

**Figure 15.** A simplified schematic of the LIFT process. (**a**) A laser is focused using a microscope objective lens with possibly optical train apertures onto the donor surface locally melting a spot on the metallic thin film. (**b**) The laser is turned off while the melt deforms the film turning into (**c**) a metal droplet which drops to a substrate below.

Several system and film parameters effect the size and morphology of the solidified droplets. Whether metal can be transferred from the donor depends on the laser energy and the thickness of the metal film as well as reflectivity of a metal at specific wavelength. Bohandy et al. found Cu is transferred at laser energies as low as 60 mJ at a film thickness of 0.41 μm and a pulse length of 15 ns. They also showed transfer with films as thick as 1.2 μm though higher laser energies were required. The diameter of the deposited metal varied with energy; however, for the most part the size was around 50 μm. In their experiment, Cu lines 50 μm in width were observed. The solidified droplets were observed to be nonuniform with significant splatter observed on the substrate. An example of the changes to droplet morphology due to laser energy or fluence is shown in Figure 16. These experiments were conducted in a vacuum (0.13 mPa) with an excimer laser (λ = 192 nm) focused by a quartz cylindrical lens (63.5 mm focal length). The distance between donor and receiver of more than 13 μm [156]. In a later series of experiments done by the same group, a neodymium doped yttrium-aluminum-garnet (Nd:YAG) laser (λ = 532 nm) was used with a microscope objective lens for Cu, Al, Ag, and Au droplet generation in air [157]. Metal droplets were formed at lower energies in air. Splatter was again found at all energies (0.05, 0.1, 0.2, and 2.0 mJ), film thicknesses (0.2 and 0.6 μm), and metals. Splatter

was reduced for lower energies and/or film thicknesses. The donor to receiver distance was less than 13 μm and pulse length was between 10 and 15 ns.

To reduce splatter and improve resolution of the final print, the donor receiver distance was kept small, from near-contact up to several tens of micrometers. Additionally, laser pulse lengths were kept short. In the work by Bohandy, this distance was never more than 13 μm. Zergioti et al. studied deposition of chromium (Cr) lines onto a receiver with varying film thicknesses (40, 80, and 200 nm) and variable donor–receiver distances (near contact up to 1000 μm ± 5 μm). The Cr donor was sputter coated (or electron-beam evaporated) onto a transparent quartz wafer. The donor was then placed in a low-vacuum (10 Pa) chamber above a Corning type 7059 glass receiver. They found higher definition with shorter donor–receiver distances (<10 μm). They also found lower splatter of the deposited lines with thinner metal films and shorter pulses (femtosecond as opposed to picoseconds or nanoseconds) [158,159]. Shorter pulses were shown to reduce thermal diffusion of the metal species [160]. Papakonstantinou et al. confirmed shorter pulses (<1 ps) lead to higher definition in deposited metal (Pt and Cr). It was also found that better prints were obtained in a vacuum and with laser fluences close to the threshold value necessary to melt the material [161]. Additional experiments with changes to the laser source (changing λ), laser fluence, and other parameters were investigated on Au, Ag, Cu, and Ni [162–164]. Other metals and metalloids printed by the LIFT method includes germanium (Ge) [165,166], palladium (Pd) [167], Sn [165], titanium (Ti) [165,168], vanadium (V) [165], and W [169].

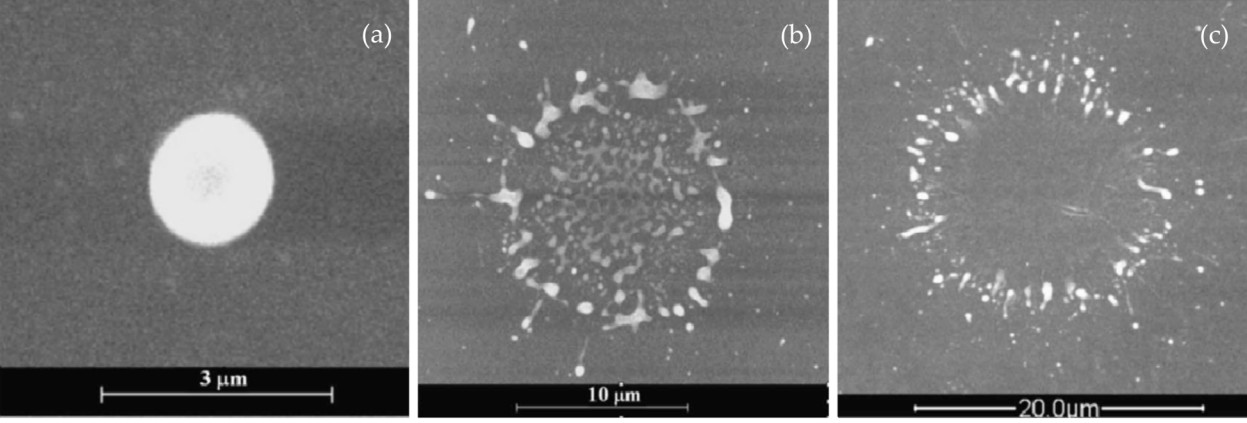

**Figure 16.** Solidified Cu droplets on a quartz substrate. (**a**) Uniform droplet observed with 6.5 μJ; (**b**) splattered droplet seen at 11 μJ laser energy; and (**c**) splattered droplet without a center region seen with an energy of 16 μJ [164]. Reprinted from Applied Physics Letters, vol. 89, issue 24, L. Yang, C. Wang, X. Ni, Z. Wang, W. Jia, and L. Chai, "Microdroplet deposition of Cu film by femtosecond laser-induced forward transfer", pp. 161110; (2006); with permission from AIP Publishing.

The mechanisms for microdroplet deposition was studied by Willis and Grosu who looked at the deposition of Al and nickel (Ni) with a 7 ns pulse Nd:YAG laser (λ = 1.064 μm) [170,171]. Film thickness was 1 μm, donor–receiver distance was from contact up to 25 μm, and laser fluence was changed. They found splatter and the size of deposited drops were larger than the laser spot size when the laser fluence was higher than the threshold value. The degree of splatter increased, and the size of the deposited drops also increased with an increase in laser fluence. A qualitative model is seen in Figure 17 [171]. At fluences near the threshold value, the authors developed a qualitative model for droplet formation. At laser fluences below the threshold small bumps appear in the donor on the metal film side. At the threshold value, small protrusions form at the central region of a bump. At values just above the threshold, holes appear in the protrusions where droplets form and leave the donor. Melting begins at the interface between the metal film and transparent substrate, a constrained surface. This means film expansion is gradual until the melt reaches the free surface of the metal film. At laser fluences much

higher than the threshold value, other mechanisms are in effect and high degrees of splatter are observed.

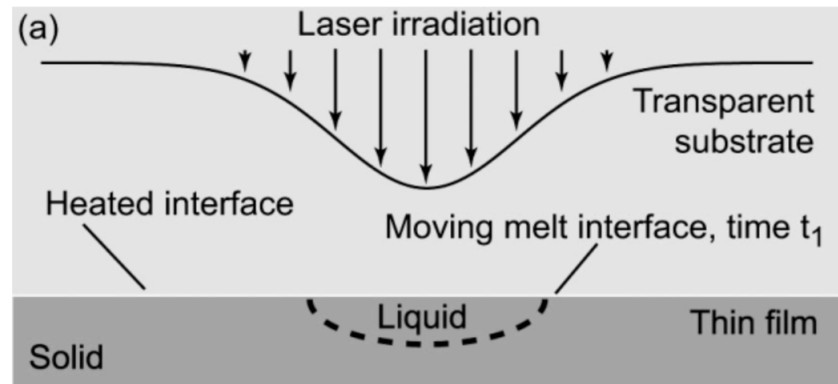

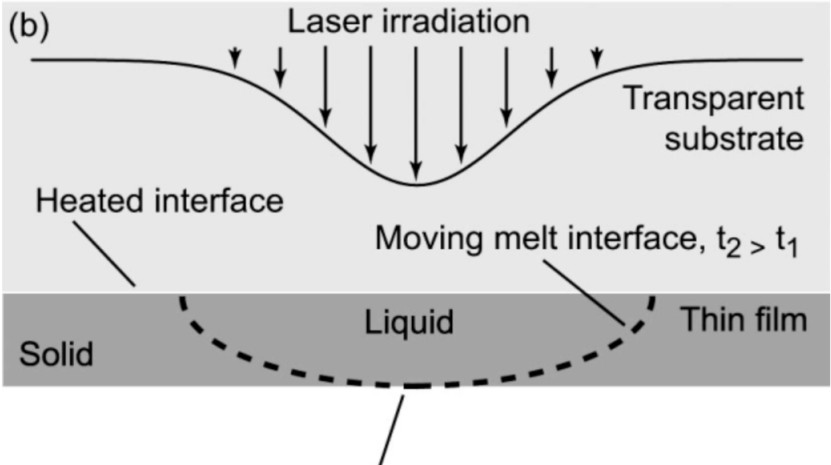

**Figure 17.** Diagram of the model of laser heating of donor films at fluences near the threshold value. (**a**) Starting laser exposure, a liquid pool is initially formed and constrained. (**b**) At some later time, the liquid metal pool has expanded and reached the free surface of the metal film [171]. Reprinted from Applied Physics Letters, vol. 86, issue 24, D. A. Willis and V. Grosu, "Microdroplet deposition by laser-induced forward transfer", pp. 244103; (2005); with permission from AIP Publishing.

Cu and Cr droplets were shown to be uniform and tear-shaped with femtosecond pulses and lower laser fluences and energies. Yang et al. showed uniform Cu droplets produced with ≈150 fs pulses of a 775 nm laser. At energies around 6.5 μJ, a uniform solid drop formed on the receiver surface. Above this laser energy produced a splattered droplet [164]. Banks et al. demonstrated sub-micron sized uniform and repeatable deposition of Cr droplets using the LIFT technique with a femtosecond Ti doped sapphire laser. They observed Cr droplets were generated even from previously irradiated portions of the donor metal film without loss of droplet resolution [172]. The volume of molten metal controls the size of the solidified droplet found on the receiver. Arrays of spherical Au nanodroplets were LIFT deposited by femtosecond laser pulses. The shape and size of the solid deposits were controlled by controlling the volume of the Au film which was liquified [173].

Printing larger objects through increasing the donor to receiver distance has also been actively investigated. The low separation between donor and receiver surfaces (<200 μm) limited the directionality of the droplets. This lack of directionality limits the LIFT technique to sub-micron sized builds, e.g., solder bumps, and limited print accuracy necessitating

the low gap. A series of works by Zenou et al. showed the printing of Al micro-droplets using donor–receiver gaps greater than 200 µm. They found Al droplets that maintained shape with little splatter or spreading on the receiver surface. The gap was varied from 10 to 300 µm using sub-nanosecond pulses of 400 ps and laser energy of 30 µJ (fluence of 0.67 J/cm$^2$). The higher laser energies were coupled with thicker metal films (up to 500 nm) [174,175]. Higher laser energies with thicker metal films causes the formation of a "nozzle" for the further jetting of molten metal. This film nozzle was also shown to resolve the issue with limited directionality by allowing for increased donor–receiver gaps. In subsequent works, Zenou et al. printed objects made of Cu/Al, Cu, Ag, or Au with this "thermal induced nozzle—LIFT" or TIN-LIFT method [176–178]. The TIN-LIFT method has resulted in fabrication of metal structures in the millimeter scale with droplet resolution where droplet sizes were near sub-µm. The structures can be complicated with various feature sizes printed at once or simple with high aspect ratios, e.g., tall and thin cylinders.

Laser-induced droplet generation has produced droplet arrays with dimensions in the sub-micron regime exemplified by the work of D.P. Banks et al. who deposited Cr droplets on Si wafers with diameters around 330 nm. They were able to create arrays of similar sized droplets using laser fluences near the threshold [172]. Objects several hundred microns in size have been printed with the LIFT technique. Through TIN-LIFT, Zenou et al. was able to print the word "small" standing for the journal Small using Cu droplets (approximately 5 µm in diameter) deposited onto a glass substrate. The size of the letters ranged from 40 µm to 190 µm in height. The width of each letter was about 40 µm indicating different aspect ratios [176]. This group and others showed the ability to use LIFT for printing of high aspect ratio objects tens to hundreds of microns in height. Metals printed to form high aspect ratio columns include Cu [177,179,180], Si [181], and Au [182]. More complex structures shown possible by these groups indicates the possibility of building larger complex metal parts via LIFT techniques especially TIN-LIFT. An example is the printing of "bent" or segmented columns where elbows are intentionally introduced to pillars of metal with the same high aspect ratios mentioned previously. Stable segmented pillars were printed with elbow angles as high as 25° [177,180].

The LIFT process resolves some issues with other DOD printing techniques, namely temperature, viscosity, and resolution limitations. As long as a metal can be deposited onto a transparent substrate, the LIFT process can be used to create droplets of that metal [178]. Therefore, the melting point of a metal or the operating temperature of the system is not as much of a concern as droplets will only contact the receiver. Viscosity and resolution, at least in terms of droplet size and volume, are also not concerns as there is no nozzle to affect the size of droplets (at least in the traditional sense) or restrict fluid flow. Several droplet formation regimes exist depending on laser energy (or fluence), pulse length, etc. Regardless of the regime, once droplets are formed, those droplets are free of the donor negating any viscosity concerns. The one exception may be droplet formation via the TIN-LIFT method where a nozzle is intentionally formed in the metal film. Further study may be needed as to any limits imposed by the TIN-LIFT method, but it appears that high-resolution free-standing parts at length scales above the micron region are possible. High-resolution prints are achievable with liquid metal printing. What follows is a discussion of the various liquid metal printing techniques in terms of part resolution and temperature limitations. The final section is the conclusion.

## 9. Discussion

### 9.1. Summary and Discussion of Liquid Metal Printing

Multiple inkjet printing technologies have been developed to additively manufacture metal objects. These techniques jet liquid in one of two modes, CIJ and DOD. The major difference between the two modes is frequency of droplet generation. In CIJ, droplets are jetted continuously. The stream can be broken up into individual droplets either through application of ultrasonic vibrations caused by a vibrating piezoelectric element or by electrostatic charging. Metal objects are built-up from streams of droplets, which are deflected

by charged plates, following a digital pattern, down to the substrate. In DOD systems, individual drops are generated, and metal objects are built-up by the sequential jetting of those individual droplets. Droplet generation rates between 10,000 and 25,000 droplets/s are seen in CIJ [112]. This is a much higher generation rate as compared to DOD which means shorter build times. The downside is increased waste and/or complexity in CIJ printing systems versus DOD. A gutter or other means of capturing unused droplets must be devised to reduce waste. As molten metal is being dealt with, the gutter must incorporate a heating system to prevent clogging. The alloys jetted by CIJ were listed in Table 1. Some of the metals which have been jetted by the DOD printing techniques are listed in Table 2.

**Table 1.** Table of liquid metal and metal alloys jetted by CIJ printers. The table is ordered in terms of the melting point ($T_m$) of the alloys from lowest to highest. The maximum operating temperature ($T_{max}$) is the maximum temperature of the generator used.

| Metal | Composition | $T_m$ (°C) | Droplet Separation | $T_{max}$ (°C) | Ref. |
|---|---|---|---|---|---|
| Solder | $Sn_{63}Pb_{37}$ | 183 | Ultrasonic vibration | 1100 | [114] |
| Sn | Sn | 232 | Electrostatic | 400 | [113] |
| Zn-Sn | $Zn_{80}Sn_{20}$ | 376 | Electrostatic | 400 | [113] |
| Al | 2024 Al | 500 | Ultrasonic vibration | 1000 | [106] |
| Al | 1100 Al | 660 | Ultrasonic vibration | 1000 | [106] |

**Table 2.** Table of liquid metal and metal alloys jetted by different droplet generators and printers. The table is ordered in terms of the melting point ($T_m$) of the alloys from lowest to highest. The maximum operating temperature ($T_{max}$) is the maximum temperature of the generator used.

| Metal | Composition | $T_m$ (°C) | Printing Technique | $T_{max}$ (°C) | Ref. |
|---|---|---|---|---|---|
| EGaIn | $Ga_{75}In_{25}$ | 15.7 | Pneumatic [1] | — | [142] |
| Hg | Hg | 23 | Electromagnetic | — | [141] |
| Bi, In | $Bi_{35}In_{48.6}Sn_{15.9}Zn_{0.4}$ | 58 | Liquid phase pneumatic | 300 | [144,145] |
| Pb-free solder | $Bi_{32.5}In_{51}Sn_{16.5}$ | 62 | EHD | 390 | [127] |
| Indalloy-158 | $Bi_{50}Pb_{26.7}Sn_{13.3}Cd_{10}$ | 70 | Piezoelectric squeeze mode | 150 | [97] |
| Pb-free solder | $Bi_{58}Sn_{42}$ | 138 | Piezoelectric push mode | 400 | [100,101] |
| Solder | $Sn_{63}Pb_{37}$ | 183 | Piezoelectric push mode | 300 | [98] |
| Solder | $Sn_{63}Pb_{37}$ | 183 | Piezoelectric squeeze mode | 600 | [102] |
| Solder | $Sn_{60}Pb_{40}$ | 190 | Impact-driven | 1230 | [155] |
| Solder | $Sn_{60}Pb_{40}$ | 190 | Starjet—Pneumatic | 280 | [148] |
| Pb-free solder | $Sn_{95}Ag_{4.0}Cu_{1.0}$ | 213 | Starjet—Pneumatic | 320 | [149] |
| Pb-free solder | $Cu_{50}Ag_{30}Sn_{20}$ | 220 | Piezoelectric push mode | 300 | [98] |
| Pb-free solder | $Sn_{99.25}Cu_{0.0075}$ | 230 | Piezoelectric push mode | 300 | [98] |
| Sn | Sn | 232 | MHD—Metaljet | 1000 | [10] |
| Solder | Unidentified | 290 | Electromagnetic | — | [141] |
| ZAMAK | $Zn_{96}Al_{4.0}$ | 420 | Starjet—Pneumatic | 500 | [149] |
| Al | 7075 Al | 480 | MHD—Magnetojet | 950 | [139] |
| Al | $AlSi_{12}$ | 577 | Pneumatic | 750 | [183] |
| Al | $AlSi_{12}$ | 577 | Starjet—Pneumatic | 950 | [184] |
| Al | 6061 Al | 585 | MHD—Magnetojet | 950 | [135,139] |
| Al | 4043 Al | 620 | MHD—Magnetojet | 950 | [139] |
| Ag | Ag | 962 | MHD—Metaljet | 1000 | [10] |
| Cu | Cu | 1080 | Pneumatic | 1100 | [151,152] |
| 52,100 Steel | $FeCr_{1.5}C_{0.1}$ | 1560 | Pneumatic | 1600 | [153,154] |

[1] Free-standing objects of EGaIn were printed by two methods, one of which did not use a gas source in jetting metal droplets.

CIJ printing using molten metal was developed for both solder deployment and freeform fabrication of metal objects. Piezoelectric driven DOD printers were developed for the printing of solders for microelectronics applications. Solders have also been printed by electrohydrodynamic printers. Magnetohydrodynamic jetting was developed for the inkjet printing of free-standing Al objects and has been demonstrated for several Al alloys in addition to Sn and Ag. Pneumatic methods of droplet generations may have the widest utility in terms of metal printability as various techniques using gas to push liquid metal out a nozzle or orifice has been shown to jet metal alloys with melting points as low as

15 °C and as high as 1560 °C. The impact-driven technique was not invented for printing with a specific metal but to print smaller metal droplets. Not included in Table 2 are metals printed by laser-induced droplet jetting. This unique technique does not have the same temperature limitations as the other DOD (and CIJ) techniques. The laser and optics used to generate metal droplets are not in thermal contact with the metal source so the only temperature concern is whether the substrate will disintegrate when a droplet hits it.

Temperature limits exist for liquid metal printers. Piezoelectric-driven techniques will not jet metals with melting points higher than the Curie point of the piezoelectric as the element will exhibit non-piezoelectric phases at higher temperatures. Insulating the piezoelectric element; however, has been shown to increase the operating temperature. For CIJ, EHD, MHD, pneumatic DOD, and impact-driven DOD the materials used to construct the printer limits the operating temperature. Liquid metal must be contained within a furnace or reservoir for melting before ejection out the nozzle. Therefore, the walls of the reservoir and the nozzle at a minimum should be stable at the melting point of the liquid metal being jetted. These concerns do not apply to laser-driven DOD. The substrate where metal objects are fabricated must be stable at elevated temperatures including in laser DOD. The distance between the nozzle and the substrate will need to be adjusted to account for the cooling and solidification of droplets as they fall to the substrate. If the nozzle and substrate are too close, then droplets may not experience sufficient cooling before impacting the substrate. This is a problem if the substrate is made of a material with a lower melting or decomposition temperature. If the nozzle and substrate are separated by a large distance, droplets will see enough cooling to solidify before impact. If droplets solidify before fusing together, then a 3D object will not be printed.

The temperature gradients and cooling rates seen in liquid metal printing are not as extreme as in LPBF or EPBF processes. As there is a small separation between the nozzle and the substrate, there is time for the droplets to cool down before impact. The separation is adjusted depending on the temperature of the melt. Upon impact, the droplets solidify as would be seen in traditional fabrication methods, i.e., typical casting or forging of metal parts. The smaller temperature gradients and cooling rates potentially mean less defects encountered such as poor consolidation and porosity as compared to LPBF or EPBF. For each technique, however, optimization would still need to be made for the best combination of, among other things, cost, resolution, and build time. Parameters to be adjusted include temperature, nozzle size, and nozzle-substrate separation distance. Post processing would also need to be performed to obtain the desired properties out of the metal parts. Additional considerations for printing liquid metals include the oxidation potential of the alloy to be printed. This problem is resolved by performing printing operations in low oxygen environments or shrouding the reservoir, driving mechanisms, nozzle, and other regions where the liquid metal is found in an inert gas like Ar.

The resolution of a build is affected by, among other things, the droplet size. Factors affecting the droplet size include the size or diameter of the nozzle, the speed of droplet jetting, and whether a single jet contains one droplet or multiple droplets. Figure 18 shows the average droplet diameter as compared to the nozzle diameter within different DOD techniques. Not included in Figure 18 are the CIJ or LIFT methods. In CIJ, droplets jetted from nozzles with a size of 100 μm were shown to have an average diameter of 190 μm [106,114]. For the DOD methods, EHD and impact driven techniques had the smallest range in both nozzle diameter and droplet size because the relatively few published works on jetting of alloy droplets (EHD: [132] and impact: [155]). An important note, EHD printing achieved droplet sizes which were 1/3 the size of the nozzle. Pneumatic DOD has the widest variation in droplet sizes, but also had the widest variation in nozzle sizes among the liquid metal printers. Piezoelectric driven DOD achieved the smallest droplet sizes. Piezoelectric push-mode printers jetted droplets as large as 450 μm and as small as 13 μm [98]. Piezoelectric DOD squeeze-mode printers jettisoned solder droplets as small as 25 μm [97]. Nozzle size has a large influence on the droplet size. Other factors affecting the droplet size depends on the jetting technique.

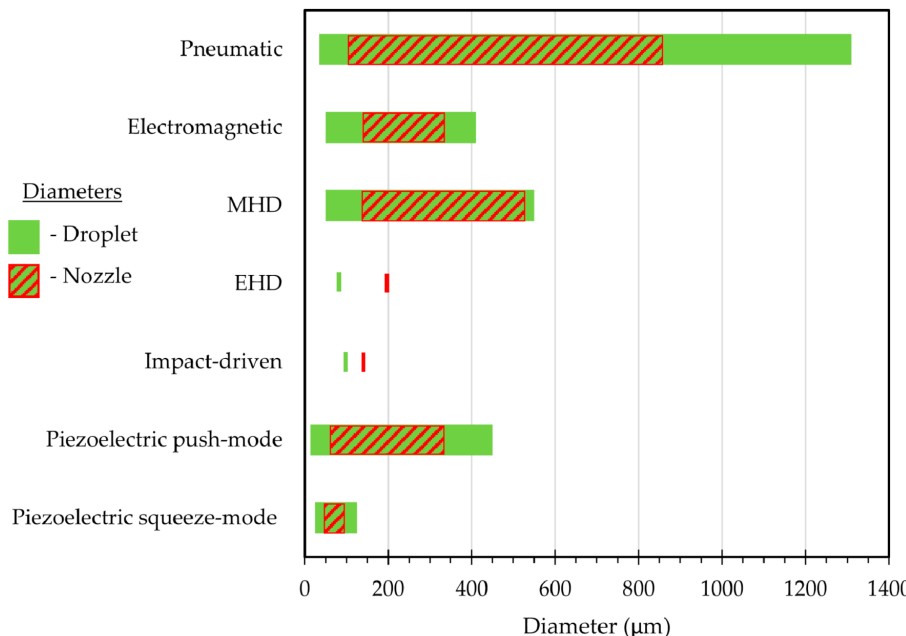

**Figure 18.** Comparison of average droplet diameters (μm) to nozzle diameter (μm) for DOD printing techniques. This does not include statistical variations. Data to make this figure drawn from multiple sources [10,97,98,100–102,106,114,127,138–141,144,148,149,152,154,183,184].

Jiang et al. found that the vibration frequency affects the droplet size and shape during formation. They also found that gas pressure and environmental effects also changes droplet morphology. Studying these parameters on $Sn_{63}$-$Pb_{37}$ jetted by a CIJ/ultrasonic printer, they found that the stream could not be broken up when the nozzle was 140 μm, gas pressure was 30 kPa, and the vibration frequency was 15 kHz. When adjusting these parameters; however, to 100 μm, 40 kPa, and 2.6 kHz, they were able to produce single droplets with small variation in size [114]. For the DOD printing techniques, several parameters in addition to nozzle size affected droplet size. Across the DOD methods, common features emerged. Droplet size changes with actuation (e.g., gas pressure for pneumatic printers or electric potential for EHD) duration and frequency. Oxidation potential will affect the size and shape of droplets regardless of jetting technique. The formation of an oxide skin in Ga droplets was found to assist in the solidification of spherical droplets on a substrate [142]. Oxides formed in the melt may also interrupt droplet jetting as was found for Al jetted in CIJ [106], and Al and Cu in DOD [152,185]. The distance or separation of the nozzle and substrate affects the droplet morphology landing on the substrate as well as the temperature of the substrate, two parameters important to the final build resolution.

The resolution and temperature of the droplets affects the dimensional accuracy and surface finish of parts and ultimately, the part quality. Figures 8 and 10 show free-standing parts printed via the EHD and MHD methods, respectively. The printed parts showed surface roughness in the sub-millimeter scale. Simonelli et al., in their work on the MetalJet system (an MHD method), reported a 0.3% up to 11% part fidelity with the CAD design. This difference in dimensional accuracy was due, in part, to the metal jetted and the operating temperature [10]. Chao et al. investigated the parameter adjustments to control part quality of droplet printed Al. They observed that the part quality improved when the temperature of droplets exiting the nozzle and striking the substrate was kept between the temperatures of 450 and 700 °C [186]. These temperatures roughly corresponded to the liquidus and solidus lines for aluminum and may be different for other alloys. Other contributors include droplet velocity, droplet-substrate interface behavior (i.e., whether splatting occurs and the height of splats) [10], and the scan speed and droplet overlapping (studied by Qi et al.) [187,188].

*9.2. Future Direction of Liquid Metal Printing*

Liquid metal printing of free-standing digital objects is not a mature technology when compared to LPBF and other beam-based AM printing techniques. To date, the only liquid metal jet printer near-commercially available are Vader Systems (now Xerox) liquid jet printers. This is compared to several LPBF manufacturers producing multiple printing platforms such as EOS, Renishaw, 3D Systems, and others. One factor for this are the temperature limitations and precursor materials. In beam-based techniques, a powder or wire is directly melted by an energy beam and following a digital pattern, a part is fabricated on a build plate. Therefore, only the precursor material is in direct contact with molten (or high temperature) material. In liquid metal printer, molten metal is in contact with the walls of the reservoir, nozzle assembly, and substrate. All these surfaces must be able to withstand high temperatures. The precursor materials for liquid metal printing may also be a concern. In LPBF and EPBF, preprocessed powder ready for printing are typically used. In liquid metal printing, the alloy must be heated and melted in situ and since these techniques are basically an advanced form of casting, consideration to impurities and oxidation in the melt must be made, i.e., include a flux for certain alloys. Two potential research directions include investigations in jetting higher melting point alloys and printing fully functional metal parts.

Developing systems which can print alloys such as steels would advance the field significantly and place liquid metal printing on with the same level as LPBF. On the second point, there have been, to date, few studies on the mechanical properties of liquid metal printed parts [106,189]. Such studies would be required adoption of fully functional liquid metal printed parts in real world applications. Another direction would be to investigate printing of nonweldable alloys, which were briefly discussed in Section 2.1. Some Al alloys and many Ni-based superalloys are non-weldable meaning printing of these via LPBF or EPBF is difficult without parts experiencing cracking or formation of pores. Liquid metal printing may potentially resolve these issues. Researchers, however, must show that cooling rates of the molten droplets upon the substrate and subsequent droplet-on-droplet cooling rates are low enough for sufficient droplet fusion and mitigation of cracking. It may be that as-printed parts will need post-processing such as heat treatment for full densification of these parts while the printing phase fabricates a green body durable enough for transport to a furnace.

A final note will be made here on the future directions liquid metal printing may take in terms of applications. The only direct application where liquid metal printing has an advantage over other techniques is in solder application on microelectronics boards. Recent work; however, performed by Yi et al. and others showed how to use liquid metal printing to fabricate microwave devices [190–192]. They used a piezoelectric element attached to a vibrating rod to jet molten aluminum micro-droplets out of an orifice. The first layer of droplets was deposited on a rotating soluble substrate. Once the core was fully coated, it was removed chemically (the core was water-soluble). Additional layers could then be deposited on the initial core. More work is needed to smooth out the inner surfaces; however, their work could lead to AM fabrication of complex microwave devices. More importantly, it shows that a focus on potential applications of liquid metal printing is the way forward.

## 10. Conclusions

This review broadly covered liquid metal printing, a family of metal jetting techniques for the AM fabrication of metal objects. Liquid metal printing is based on thermal and piezoelectric driven inkjet printers, which have been commercially available for over half a century. Liquid metal printing, in general, involves the direct melting of alloys; jetting of the molten metal as droplets; and building of 3D objects following a digital pattern. There are different techniques with different droplet generation and jetting methods. CIJ jets molten metal out as a stream and droplets are generated either by applying an electric field across the stream or using a piezoelectric to generate ultrasonic vibrations. DOD printing

is distinguished from CIJ as the direct jetting of individual droplets as opposed to a stream of fluid broken into individual droplets. The several DOD techniques generate and jet metal droplets in different ways. Reviewed DOD techniques included piezoelectric-driven, EHD, MHD, pneumatic, impact, and laser-driven DOD. Regardless of the method, for part resolution, the parameters of nozzle size, pulse frequency, and pulse duration need to be considered. All methods discussed have temperature requirements due to the construction of the printer and melting point of the targeted alloy. Other advantages and areas requiring more research were identified. Liquid metal printing could resolve some issues with other AM techniques and could be used for the fabrication of unique metal structures.

**Author Contributions:** Troy Y. Ansell researched and wrote the entire article. The author has read and agreed to the published version of the manuscript.

**Funding:** This research received no external funding.

**Institutional Review Board Statement:** Not applicable.

**Informed Consent Statement:** Not applicable.

**Data Availability Statement:** No new data were created or analyzed in this study. Data sharing is not applicable to this article.

**Conflicts of Interest:** I declare no conflict of interest in writing this review. The views expressed in this document are those of the author and do not reflect the official policy or position of the Department of Defense or the U.S. Government.

## Abbreviations

| Additive manufacturing | AM |
|---|---|
| Drop-on-demand | DOD |
| Electrohydrodynamic | EHD |
| Magnetohydrodynamic | MHD |
| Computer-aided design | CAD |
| Computer-aided manufacturing | CAM |
| Continuous inkjet printing | CIJ |
| Fused filament fabrication | FFF |
| Selective laser sintering | SLS |
| Selective laser melting | SLM |
| Laser powder bed fusion | LPBF |
| Electron powder bed fusion | EPBF |
| Lead zirconate titanate | PZT |
| Polytetrafluoroethylene | PTFE |
| Laser induced forward transfer | LIFT |
| Thermal induced nozzle | TIN |

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
