# Peer review of "Current Status of Liquid Metal Printing"

_jmmp, doi:10.3390/jmmp5020031_

Round 1
Reviewer 1 Report
In the paper Current Status of Liquid Metal Printing there are few details, presented below, which require further attention by the author.
Comments and suggestions
Affiliation: Please provide complete address information (see Instructions for Authors: https://www.mdpi.com/journal/jmmp/instructions)
Page 4
Line 111: An explanation of the abbreviation DLP is necessary
Line 122: Please briefly explain the difference between SLS and SLM (to eliminate confusion for readers). Even if the principle of deposition is similar, there are differences between them.
Author Response
I want to thank the reviewer for their comments on my manuscript. Below are my responses to their comments.
- My affiliation has been corrected with my full address.
- A brief explanation on DLP has been added.
- Brief explanations of both the SLS and SLM techniques has been added.
Thank you again.
- Troy Ansell
Reviewer 2 Report
This review focuses on the current state of the art of liquid metal additive manufacturing. The continues and DOD has been discussed. However, the few comments below seem to be essential to be taken into consideration before we proceed with this publication:
-citation to the recent articles in AM has not been discussed. essential to reference to: https://doi.org/10.1016/j.promfg.2020.04.127
-the difference of SLS and SLM has not been well described. a diagram or schematic in this would really help to understand this faster.
-advantages of higher operating temperatures in DOD should have been compared with continues methods. line 454 only mentioned briefly.
-Shear mode piezoelectric actuators for inkjet applications requires a comparison with the conventional methods.
when authors take my comments i ca reconsider my decision.
Author Response
I want to thank you for taking the time to review this manuscript and for your comments on improving this review paper. The following are my responses to your comments.
- This reference was added and is found in lines 64 - 68.
- I think this paper has enough figures in its current form. I did add a short description on SLS and SLM found on lines 138 and 139. I would like to keep the description short as this paper is about liquid metal printing.
- It is mentioned that common alloys such as steels would obviously require a system with higher operating temperatures, in this case lines 785 - 809. I compare the current temperature limits of the techniques through tables 1 and 2. There are advantages of jetting alloys at temperatures higher than the melting point, but this is along one technique. Each technique could conceivably jet high melting point alloys like steel, the only limiting factor seems to be the materials used in building the printer. I focus on these temperature limits as they currently stand.
- An explanation of how a piezoelectric shear mode inkjet printhead is provided on lines 312 - 319. A further brief explanation for why we do not see this type of jetting for molten metal is provided on lines 486 - 488. As far as know, no one has developed a shear mode printer of a purely metal fluid. I would imagine this type of jetting technique would be ideal for micro-(or nano-)droplet generation. It would; however, be severely restricted by temperature limits. One sentence is added to lines 488 - 490 to explain.
Thank you again for your comments and I hope my answers to your comments are sufficient.
-Troy Ansell
Reviewer 3 Report
This article is original and a great contribution to AM community. This reviewer also checked the originality of this article through Turnitin. The Turnitin system shows that this article is novel. However, this reviewer still has the following concerns:
Liquid metal additive manufacturing could not be labeled as AM.
For figure 1, have image for each item. Or, have no image for each item.
There is no reference or base for Figure 2.
Figure 15 has no impact for the review paper.
Label Figure 16.
Table 2 is acceptable. However, there is pros and cons discussion of the printing techniques in Table 2. Why?
Section 2 is the background. Section 3 is the Continuous Liquid Metal Printing and Section 4 is the Piezoelectric Drop-on-Demand Printing. After Section is made, there should be a classification section detailing the sections 3 and 4. The flow is hard to follow.
There is good reporting on the parameters in dimensional accuracy, surface finish and part quality of the parts made with liquid metal additive manufacturing. Details and additions are needed.
Figure 1 is for the generic AM technologies. The reader is more interested to see such a flowchart for the liquid metal additive manufacturing. This is important.
Lately, there are papers published on low cost metal 3D printing by Gong, Fidan and many others. Benchmarking the liquid metal additive manufacturing to low cost metal additive manufacturing could be a great contribution presented in this article.
Author Response
I want to thank the reviewer for taking the time to make comments and suggestions in order to improve this manuscript. I attempt to address all concerns raised in this manuscript and list them below in the order the reviewer listed their comments.
- It is true that on the face of things, liquid metal printing sounds like a variant of casting. Whether or not liquid metal printing could be classified as an AM technique probably depends on the user. I believe this jetting technique can be considered an AM technique because a 3D free standing part could be fabricated using the techniques discussed in this paper. Whether post printing polishing and/or machining is required will depend on the print resolution. To my knowledge, no one has looked into the finer details of this. Only into the ability of a printing technique to print parts. Ultimately, liquid metal printing falls under material jetting defined by ISO / ASTM52900-15 as AM. So, I feel liquid metal printing is an AM technique.
- I have changed figure 1 to address this concern, there are no figures just words.
- A reference is added with a short explanation to the Fig. 2 caption.
- Figure 15 provides to the reader, a schematic for the laser induced DOD technique. I feel this figure should be included to give the reader a visual for how the technique works in general. Once given this visual, the reader could go to the references to get a detailed take on adjustments to the technique or specific parameters.
- Figure 16 has a caption and is referenced in the text, line 885.
- I used table 2 primarily to contrast the upper temperature limits of the different techniques. I also used the table to start a discussion on other aspects of liquid metal printing, namely the resolution of builds.
- The end of section 2 with an explanation for the following sections has been updated to address your concern on the paper's flow.
- I do indirectly touch on the topics of dimensional accuracy, surface finish, and part quality through a discussion on resolution in section 9. I however, expand on this to directly touch on part quality. This can be found at the end of section 9.1 in the revised manuscript.
- As stated in an earlier response, figure 1 was changed. The change made also addresses the concern regarding reader interest. The timeline was changed to focus only on liquid metal printing's history.
- I feel that a comparison between metal printing via one of the material extrusion methods discussed in the papers you cite or by FDM or stereolithography is beyond the scope of this paper. This is partly due to the fact that those figures are often difficult to obtain. Liquid metal printing is only now becoming commercially available and all indications are that the technique will be as costly as LPBF making it one or two orders of magnitude more expensive than FDM as an example. The family of techniques could however compete with LPBF on a production scale perhaps. This remains to be seen and I feel is beyond the scope of this paper.
I hope that I have adequately addressed all of your concerns. Again, thank you for taking the time to review my paper.
- Troy Ansell